# BUB-1 and CENP-C recruit PLK-1 to control chromosome alignment and segregation during meiosis I in *C. elegans* oocytes

Samuel JP Taylor[1†], Laura Bel Borja[1†], Flavie Soubigou[1], Jack Houston[2], Dhanya K Cheerambathur[3], Federico Pelisch[1*]

[1]Centre for Gene Regulation and Expression, Sir James Black Centre, School of Life Sciences, University of Dundee, Dundee, United Kingdom; [2]Ludwig Institute for Cancer Research, San Diego Branch, La Jolla, United States; [3]Wellcome Centre for Cell Biology & Institute of Cell Biology, School of Biological Sciences, University of Edinburgh, Edinburgh, United Kingdom

**Abstract** Phosphorylation is a key post-translational modification that is utilised in many biological processes for the rapid and reversible regulation of protein localisation and activity. Polo-like kinase 1 (PLK-1) is essential for both mitotic and meiotic cell divisions, with key functions being conserved in eukaryotes. The roles and regulation of PLK-1 during mitosis have been well characterised. However, the discrete roles and regulation of PLK-1 during meiosis have remained obscure. Here, we used *Caenorhabditis elegans* oocytes to show that PLK-1 plays distinct roles in meiotic spindle assembly and/or stability, chromosome alignment and segregation, and polar body extrusion during meiosis I. Furthermore, by a combination of live imaging and biochemical analysis we identified the chromosomal recruitment mechanisms of PLK-1 during *C. elegans* oocyte meiosis. The spindle assembly checkpoint kinase BUB-1 directly recruits PLK-1 to the kinetochore and midbivalent while the chromosome arm population of PLK-1 depends on a direct interaction with the centromeric-associated protein CENP-C[HCP-4]. We found that perturbing both BUB-1 and CENP-C[HCP-4] recruitment of PLK-1 leads to severe meiotic defects, resulting in highly aneuploid oocytes. Overall, our results shed light on the roles played by PLK-1 during oocyte meiosis and provide a mechanistic understanding of PLK-1 targeting to meiotic chromosomes.

**\*For correspondence:**
f.pelisch@dundee.ac.uk

†These authors contributed equally to this work

## Editor's evaluation

This study represents an important advance in our understanding of how female meiosis is regulated. By combining biochemistry with some beautiful cell biology, the authors identify and increase our understanding of how chromosome behaviour is controlled in mitosis. The data are convincing and the paper will be of interest to researchers in the mitosis and meiosis fields.

## Introduction

Oocyte meiosis consists of two consecutive segregation events following DNA replication – in meiosis I homologous chromosomes segregate, half of which are then removed in a polar body. The remaining sister chromatids segregate in meiosis II, half of which are removed in a second polar body to produce a haploid gamete (*Marston and Amon, 2004*; *Ohkura, 2015*). Tight spatial and temporal control of protein localisation and activity is required to ensure chromosome/chromatid alignment

and segregation occur efficiently and the correct number of chromosomes is present in each gamete (*Marston and Wassmann, 2017*). Phosphorylation is a key post-translational modification utilised to regulate protein localisation and activity, which is fundamentally important for the success of both mitotic and meiotic cell divisions (*Marston and Wassmann, 2017*; *Saurin, 2018*). Therefore, the regulation of kinase or phosphatase localisation and activity is vital for proper cell division, with the balance of their effects determining the localisation/activity of substrate proteins that play important roles in the cell division process (*Gelens et al., 2018*; *Novak et al., 2010*).

Polo-like kinases (PLKs) are a family of Ser/Thr protein kinases first discovered in *Drosophila* (*Llamazares et al., 1991*; *Sunkel and Glover, 1988*) and yeast (*Kitada et al., 1993*; *Ohkura et al., 1995*), and later findings showed that PLKs are present in all eukaryotes (*Zitouni et al., 2014*). PLK-1 is essential for meiotic and mitotic cell divisions, with its localisation and functions proving well conserved throughout eukaryotic evolution. PLK-1 localises to the nuclear envelope in prophase (*Linder et al., 2017*) and to the centrosomes, kinetochore, central spindle, and midbody during mitosis and is involved in numerous processes, including mitotic entry, spindle assembly, chromosome alignment, the spindle assembly checkpoint, and cytokinesis (*Archambault and Glover, 2009*; *Petronczki et al., 2008*; *Schmucker and Sumara, 2014*; *Zitouni et al., 2014*). During mammalian meiosis, PLK1 localises to the chromosomes and spindle poles in prometaphase and metaphase. In anaphase, PLK1 is localised primarily in the central spindle between the segregating chromosomes and then the midbody during polar body extrusion (*Pahlavan et al., 2000*; *Solc et al., 2015*; *Tong et al., 2002*; *Wianny et al., 1998*). Inhibition of PLK1 in mammalian oocytes has indicated roles in many processes, including germinal vesicle breakdown, spindle assembly, chromosome alignment, and polar body extrusion (*Solc et al., 2015*; *Tong et al., 2002*). To understand why PLK-1 is crucial for meiosis, it is critical to characterise precisely where PLK-1 is located and how it is recruited to specific regions, which will allow further dissection of the discrete roles of PLK-1 during meiosis.

PLK-1 interacts with proteins via its C-terminal polo-binding domain (PBD) (*Cheng, 2003*; *Elia et al., 2003a*). The PBD binds to phosphorylated motifs of the consensus Ser-phSer/phThr-X, where ph indicates a phosphorylated residue and X indicates any amino acid (*Elia et al., 2003a*; *Elia et al., 2003b*). When X is a proline, phosphorylation of the central Ser/Thr residue is often mediated by a proline-directed kinase, notably Cdk1/Cyclin B during cell division (*Elowe et al., 2007*; *Qi et al., 2006*) – this motif will henceforth be termed STP motif. When a non-proline residue occupies position X, PLK-1 itself can phosphorylate the central Ser/Thr, thereby enhancing its own recruitment (*Kang et al., 2006*; *Neef et al., 2003*), referred to as self-priming. Furthermore, PLK-1 binding to STP motifs via the PBD induces a conformational change that enhances its kinase activity (*Mundt et al., 1997*; *Xu et al., 2013*).

While the mechanism of PLK-1 recruitment to the chromosomes during oocyte meiosis has not been characterised, recruitment of PLK1 to the kinetochore during mammalian mitosis has been investigated. In mammals, the constitutive centromere-associated network (CCAN) complex of proteins binds to the histone variant CENP-A at centromeres (*Foltz et al., 2006*; *Izuta et al., 2006*; *Okada et al., 2006*). Outer kinetochore proteins bind to the CCAN and ultimately mediate chromosome alignment and segregation via interaction with microtubules (*Musacchio and Desai, 2017*). Two proteins are primarily responsible for PLK1 recruitment to the kinetochore – CCAN component CENP-U (*Kang et al., 2006*; *Kang et al., 2011*; *Singh et al., 2021*) and the spindle assembly checkpoint kinase BUB-1 (*Elowe et al., 2007*; *Qi et al., 2006*; *Singh et al., 2021*), both of which directly bind to PLK1 via STP motifs in a Cdk1-dependent manner.

In *Caenorhabditis elegans*, PLK-1 localisation in meiosis and mitosis is similar to other organisms (*Chase et al., 2000*). In mitosis, roles of PLK-1 include nuclear envelope breakdown (NEBD) (*Martino et al., 2017*), merge of parental genomes in the embryo through lamina disassembly (*Rahman et al., 2015*; *Velez-Aguilera et al., 2020*), centrosome maturation (*Cabral et al., 2019*; *Decker et al., 2011*; *Ohta et al., 2021*; *Woodruff et al., 2015*), and cytokinesis (*Gómez-Cavazos et al., 2020*) – indicating that major mitotic roles of PLK-1 are conserved in *C. elegans*. However, meiotic roles of PLK-1 in *C. elegans* oocytes have remained obscure as PLK-1 depletion results in severely defective NEBD and fertilised oocytes with a whole nucleus rather than condensed chromosomes (*Chase et al., 2000*). In the PLK-1-depleted oocytes that 'escaped' the NEBD defect, chromosome congression, segregation, and polar body extrusion were severely disrupted. However, it is unclear whether these phenotypes are the indirect effects of the severe early meiotic defects or whether they result from specific

functions of PLK-1 throughout meiosis (*Chase et al., 2000*). Furthermore, while PLK-1 was shown to localise broadly to chromosomes and the spindle during meiosis (*Chase et al., 2000*), a more precise dynamic characterisation of PLK-1 localisation during meiosis is lacking.

Here, by temporally inhibiting an analogue-sensitive PLK-1 mutant we show that PLK-1 is involved in spindle assembly and/or stability, chromosome alignment and segregation, and polar body extrusion in *C. elegans* oocytes. Using live imaging and immunofluorescence, we find that PLK-1 localises to the spindle poles, chromosome arms, kinetochores, and midbivalent region between the homologous chromosomes during meiosis I in *C. elegans* oocytes. By a combination of live imaging and in vitro biochemical analysis, we have characterised the primary chromosomal recruitment mechanisms of PLK-1 during meiosis, showing that CENP-C[HCP-4] directly recruits PLK-1 to the chromosome arms while PLK-1 recruitment to the midbivalent and kinetochore is mediated by a direct interaction with BUB-1. Furthermore, BUB-1- and CENP-C-mediated PLK-1 recruitment to chromosomes is essential for meiosis I.

## Results

### PLK-1 plays roles in spindle assembly and/or stability, chromosome alignment and segregation, and polar body extrusion in meiosis I

A previous study showed that PLK-1 localises to the meiotic spindle and chromosomes in *C. elegans* oocytes and depletion of PLK-1 using RNAi led to several defects, including defective NEBD, chromosome congression, segregation, and polar body extrusion (*Chase et al., 2000*). While this suggested that PLK-1 plays several roles during meiosis, the use of RNAi presents a limitation to addressing them independently. In particular, the strong NEBD defect complicates delineation of the roles of PLK-1 at later stages of meiosis. Therefore, we sought to understand the distinct localisation and roles of PLK-1 during meiosis I.

To assess PLK-1 localisation during meiosis with high spatial and temporal resolution, we imaged endogenously tagged PLK-1::sfGFP in dissected oocytes. PLK-1 localises to the spindle poles (*Figure 1C*, blue arrows), chromosome arms (*Figure 1A and C*, yellow arrows), and midbivalent region between the homologous chromosomes (*Figure 1A and C*, magenta arrow) during prometaphase I. This localisation pattern was confirmed with immunostaining of fixed oocytes using a specific anti-PLK-1 antibody (*Figure 1B*). As chromosomes begin to segregate in early anaphase, PLK-1 is mostly observed on chromosomes and, to a lesser extent, in between the segregating chromosome masses (*Figure 1C*, yellow and magenta arrows, respectively; see also *Figure 1—figure supplement 1*). During late anaphase, PLK-1 is still detectable on chromosomes, and is enriched in the central spindle (*Figure 1C*, green arrow; see also *Figure 1—figure supplement 1*).

Since long-term depletion of PLK-1 leads to severe NEBD defects (*Chase et al., 2000*), we used an analogue-sensitive *plk-1* allele (*Gómez-Cavazos et al., 2020*; *Woodruff et al., 2015*) that renders it sensitive to chemically modified derivatives of PP1, an Src family inhibitor (*Bishop et al., 2000*). We reasoned that acute PLK-1[as] inhibition for a short period of time would allow us to study the post-NEBD effects (*Figure 1D*). We tested a variety of analogues, and all of them led to embryonic lethality of the *plk-1[as]* strain without severely affecting a wild type strain (*Figure 1—figure supplement 2*). We decided to continue our experiments with the 3-substituted benzyl PP1 derivative 3IB-PP1 (*Figure 1—figure supplement 2*). A wild type strain in the presence of 3IB-PP1 and *plk-1[as]* in the presence of vehicle control ('EtOH') behaved normally during meiosis (*Figure 1E and F* and *Figure 1—figure supplement 3*). Addition of 10 µM 3IB-PP1 between 5 and 15 min before dissection and imaging of oocytes allowed us to bypass the NEBD defect, and six bivalents were easily identifiable within the newly fertilised oocyte (*Figure 1E*, yellow arrows). Under these conditions PLK-1[as] inhibition led to drastic spindle defects with no observable bipolar spindle formation and no consequent chromosome segregation was observed, indicating that PLK-1 is involved in spindle assembly and/or stability during oocyte meiosis (*Figure 1E*). We sought to minimise the spindle defects upon PLK-1 inhibition by reducing the concentration of 3IB-PP1 to 1 µM and omitting the pre-treatment step prior to dissection. Under these conditions, ~62% of oocytes had seemingly bipolar spindles and chromosomes remained somewhat associated with the spindle, although chromosome alignment was still affected in 78% of oocytes (≥2 misaligned chromosomes, *Figure 1F and G*). A more detailed analysis of chromosome dynamics after PLK-1 inhibition is presented in *Figure 1—figure supplement*

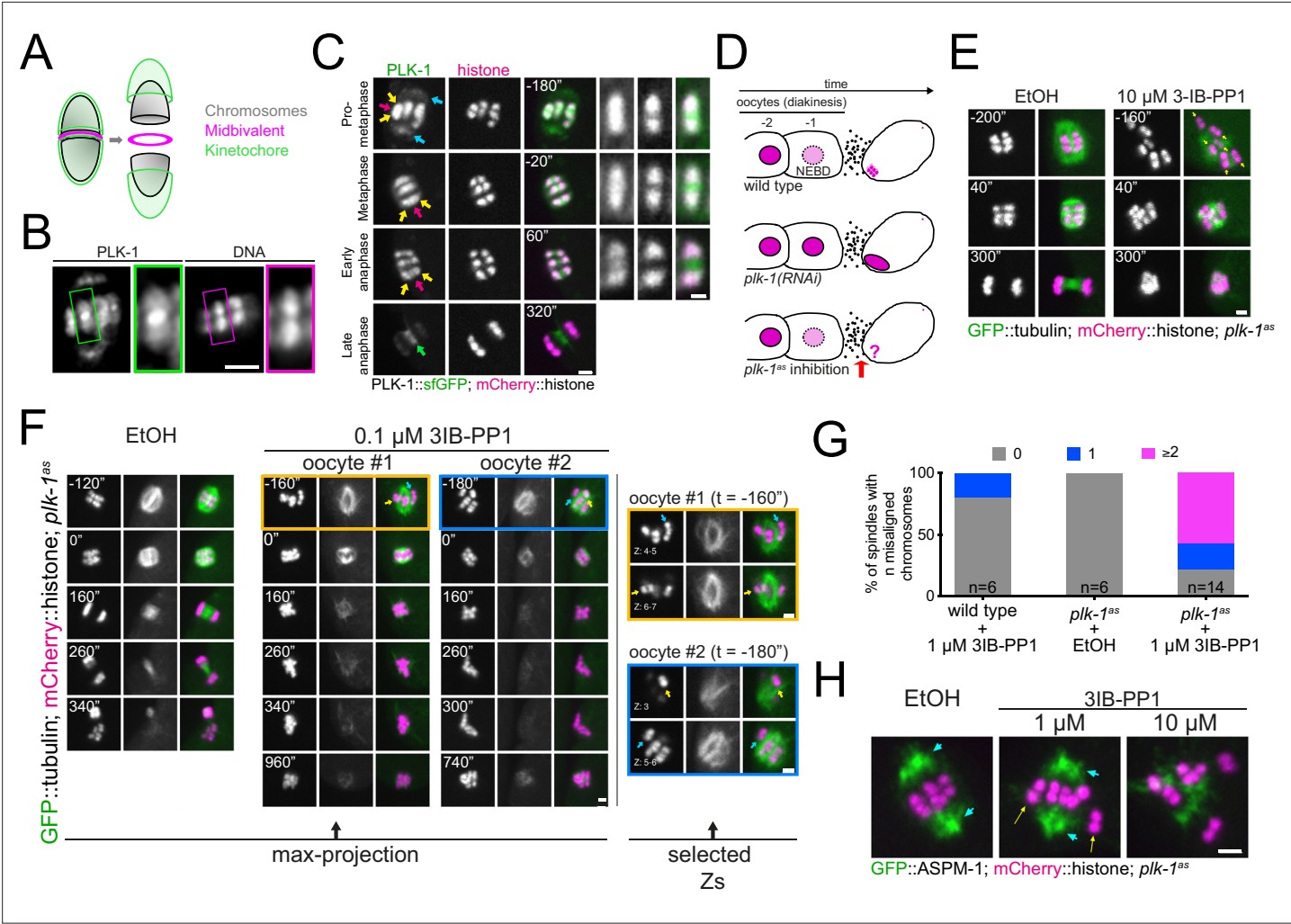

**Figure 1.** Analysis of PLK-1 localisation and inhibition during meiosis I. (**A**) Schematic of a *C. elegans* bivalent, highlighting the midbivalent and kinetochore. (**B**) Immunostaining of (untagged) PLK-1 in fixed oocytes. The insets represent a magnified image of single bivalents for each channel. Scale bar, 2 µm. (**C**) In situ GFP-tagged PLK-1 (**Martino et al., 2017**) was followed through meiosis I in live, dissected oocytes. More detailed sequence of events are displayed in **Figure 1—figure supplement 1**. Yellow arrows point to chromosomes, blue arrows indicate spindle poles, magenta arrows point towards the midbivalent, and green arrow indicates the central spindle. Scale bars, 2 µm and 1 µm (in zoomed-in bivalents). (**D**) Schematic of the last two maturing oocytes and the fertilised egg, highlighting the difference between PLK-1 depletion and acute PLK-1 inhibition. (**E**) *plk-1^as* worms expressing GFP-tagged tubulin and mCherry-tagged histone were dissected in medium containing ethanol ('EtOH', control) or the PP1 analogue 3IB-PP1 (10 µM). Yellow arrows point to the each of the six bivalents. Scale bar, 2 µm. (**F**) *plk-1^as* worms expressing GFP-tagged tubulin and mCherry-tagged histone were dissected in medium containing ethanol ('EtOH', control) or 0.1 µM PP1 analogue 3IB-PP1. Scale bar, 2 µm. The panels on the right show specific Z slices to highlight individual chromosomes. Yellow arrows point to misaligned chromosomes contained within the spindle, whereas blue arrows indicate chromosomes outside the spindle. See also **Figure 1—figure supplement 3**. (**G**) Chromosome alignment defects scored for 1 µM 3IB-PP1-treated oocytes are presented in the graph. (**H**) Worms expressing GFP-tagged ASPM-1 (pole marker) and mCherry-tagged histone along with analogue-sensitive *plk-1* were dissected in medium containing ethanol ('EtOH', control) or the PP1 analogue 3IB-PP1 at 1 µM or 10 µM. Scale bar, 2 µm. Early metaphase I spindles are shown for each condition. Cyan arrows point to spindle poles whereas yellow arrows indicate misaligned chromosomes.

The online version of this article includes the following source data and figure supplement(s) for figure 1:

**Figure supplement 1.** PLK-1 localisation during oocyte meiosis.

**Figure supplement 2.** Embryonic viability assays after PLK-1^as inhibition.

**Figure supplement 2—source data 1.** Data utilised to generate the graphs in **Figure 1—figure supplement 2**.

**Figure supplement 3.** Chromosome alignment defects upon acute PLK-1^as inhibition.

*3*, where individual chromosomes are followed every 20 s and, as opposed to wild type, chromosomes from PLK-1-inhibited oocytes show a highly dynamic behaviour whereby they seem to briefly align and then become misaligned again (*Figure 1—figure supplement 3* and arrows therein). We then used the pole marker ASPM-1 to allow proper characterisation of spindle bipolarity under these conditions and confirmed that even when two ASPM-1 poles are clearly discerned (*Figure 1H*, blue arrows), chromosome alignment fails (*Figure 1H*, yellow arrows). Hence, it appears that PLK-1 participates in chromosome alignment in a manner that is at least partially independent of its roles in overall spindle assembly and/or stability.

## BUB-1 recruits PLK-1 to the midbivalent during oocyte meiosis

To further understand the role of PLK-1 during meiosis, we sought to identify the PLK-1 recruitment mechanism(s). In mammalian mitosis, the kinase BUB-1 and its paralog BUBR1 directly recruit PLK-1 to the kinetochore via STP motifs that are phosphorylated by Cdk1 (*Elowe et al., 2007*; *Qi et al., 2006*). While the *C. elegans* BUBR1 ortholog MAD3$^{SAN-1}$ does not localise to the chromosomes or spindle during meiosis (*Bel Borja et al., 2020*), BUB-1 localises to the kinetochores and midbivalent region (*Dumont et al., 2010*; *Monen et al., 2005*; *Pelisch et al., 2019*; *Pelisch et al., 2017*). We investigated whether BUB-1 was involved in PLK-1 targeting during *C. elegans* meiosis. RNAi-mediated depletion of BUB-1 led to the loss of PLK-1 from the midbivalent (*Figure 2A and B*, blue arrows). In contrast, PLK-1 signal at chromosome arms remained unaffected (*Figure 2A and B*, yellow arrows). Analysis of the BUB-1 protein sequence revealed a putative polo-docking STP motif in amino acids 526–528 that is conserved in nematode species (*Figure 2C*). Therefore, we sought to identify whether *C. elegans* BUB-1 directly interacts with PLK-1 to mediate its recruitment to the midbivalent region.

## BUB-1 directly interacts with PLK-1 through a Cdk1-dependent STP motif

To test whether BUB-1 directly interacts with PLK-1 in vitro, we purified a recombinant fragment of BUB-1 encompassing the intrinsically disordered region between the TPR and kinase domains that contains the putative STP motif ('BUB-1$^{190-628}$', *Figure 2D*). Since the interaction between STP motifs and the PBD of PLK-1 requires phosphorylation of the central Ser/Thr residue (*Elia et al., 2003a*; *Elia et al., 2003b*), we conducted kinase assays to assess the phosphorylation of BUB-1$^{190-628}$. Cdk1 and PLK-1 can both phosphorylate BUB-1$^{190-628}$ and Cdk1 can specifically phosphorylate the STP motif at T527 (*Figure 2D and E* and *Figure 2—figure supplement 1*). Interestingly, kinase assays conducted with Cdk1 and PLK-1 together produced a prominent shifted band representing phosphorylated protein that was not present with the individual kinases (*Figure 2D*). Since STP motifs are known to be targets of proline-directed kinases such as Cdk1 (*Elowe et al., 2007*; *Qi et al., 2006*), we hypothesised that phosphorylation of T527 by Cdk1 enables PLK-1 to bind directly to BUB-1 resulting in the shifted band representing highly phosphorylated BUB-1. To test this hypothesis, we mutated T527 in the STP motif to alanine ('BUB-1$^{190-628}$(T527A)') and assessed the resulting phosphorylation (*Figure 2F*). T527A mutation in BUB-1$^{190-628}$ largely prevented the shift observed in the combined Cdk1 and PLK-1 assay (*Figure 2F*), indicating that phosphorylation of this residue is essential for the shifted band observed.

To determine whether BUB-1 directly binds to PLK-1, we purified a maltose-binding protein (MBP)-tagged PLK1 PBD (MBP-PLK1$^{PBD}$) (*Singh et al., 2021*) and incubated it with unphosphorylated or Cdk1-phosphorylated BUB-1$^{190-628}$ before assessing complex formation by size-exclusion chromatography (SEC). Cdk1-phosphorylated BUB-1 formed a stable complex with MBP-PLK1$^{PBD}$ (*Figure 3A*). SEC coupled with multi-angle light scattering (SEC-MALS) indicated the stoichiometry of this complex is 1:1 (*Figure 3B*). While unphosphorylated BUB-1 showed some interaction with MBP-PLK1$^{PBD}$, this complex eluted from the column at a higher volume and bound to a lower proportion of MBP-PLK1$^{PBD}$, suggestive of a weaker interaction (*Figure 3A*). To directly test whether phosphorylation of T527 is required for PLK-1 to bind to BUB-1, MBP-PLK1$^{PBD}$ was incubated with Cdk1-phosphorylated BUB-1$^{190-628}$ or BUB-1$^{190-628}$(T527A) and SEC was used to assess complex formation. Interestingly, when Cdk1-phosphorylated BUB-1$^{190-628}$(T527A) was incubated with MBP-PLK1$^{PBD}$ the resulting elution was reminiscent of the unphosphorylated wild type BUB-1$^{190-628}$, with a reduced proportion of MBP-PLK1$^{PBD}$ binding and elution occurring at a higher volume (*Figure 3C*). Together, these data indicate that there is a Cdk1 phosphorylation-dependent interaction between BUB-1 and PLK1$^{PBD}$ that requires phosphorylation of T527 within the STP motif of BUB-1.

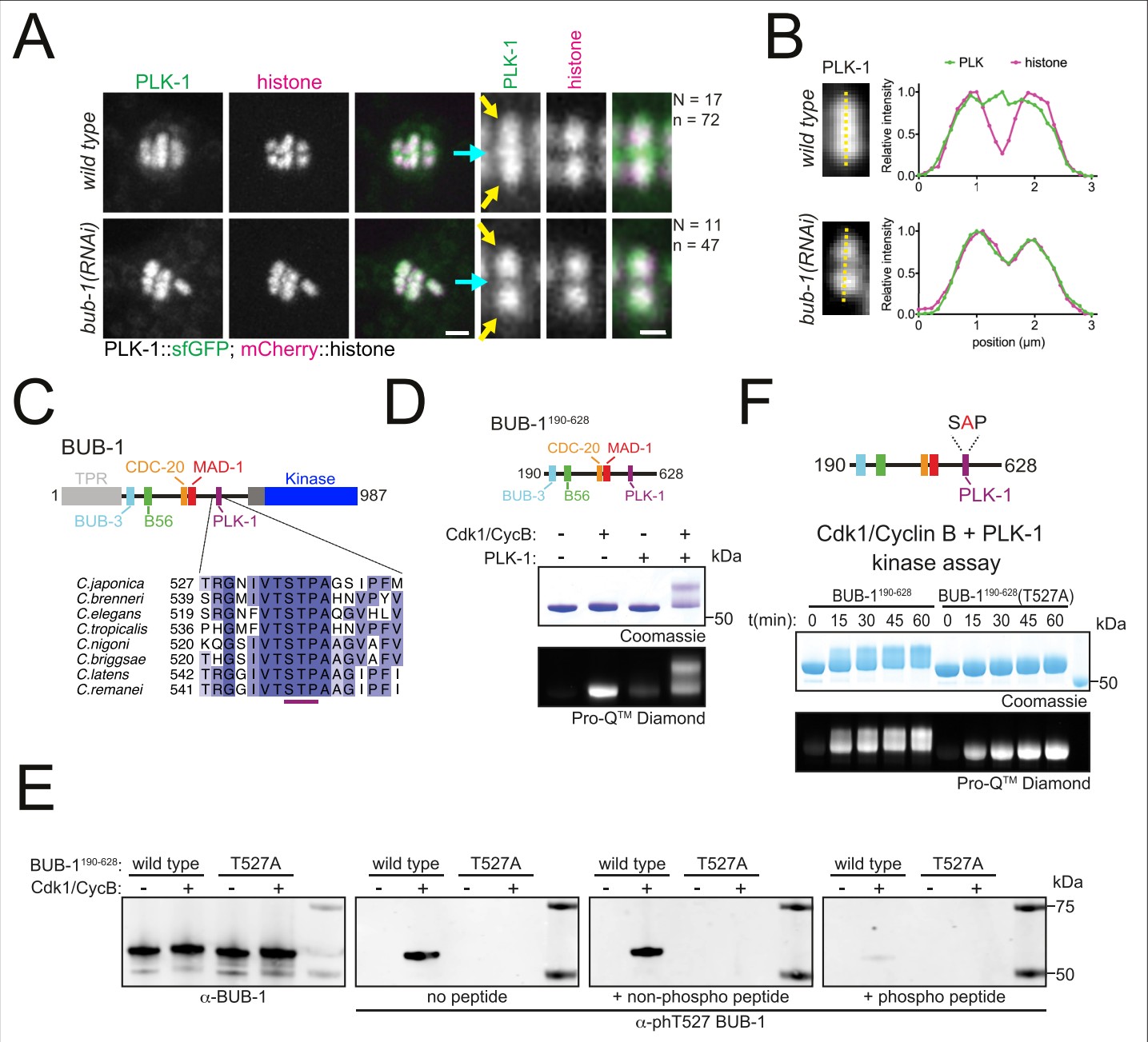

**Figure 2.** BUB-1 contains a putative STP motif and regulates PLK-1 localisation in vivo. (**A**) Control (*'wild type'*) and BUB-1-depleted [*'bub-1(RNAi)'*] oocytes expressing sfGFP-tagged PLK-1 (and mCherry-tagged histone) were dissected and recorded throughout meiosis I. Prometaphase/metaphase I is shown (before spindle rotation/shortening). The cyan arrows point to the midbivalent, and the yellow arrows point to the chromosome arms. 'N' and 'n' denote the number of oocytes and bivalents analysed, respectively. Left scale bar, 2 µm. Right scale bar, 1 µm. (**B**) Line profile analysis of PLK-1::sfGFP during prometaphase I in *wild type* and *bub-1(RNAi)* oocytes, as indicated by the yellow lines. Background signal was subtracted, and maximum signal for each channel was set to 1. (**C**) Schematic representation of the *C. elegans* BUB-1 protein (top) and sequence alignment of the putative STP motif in nematode species (bottom). (**D**) (top) Schematic of BUB-1190-628 (bottom) kinase assay of recombinant BUB-1190-628 with Cdk1/Cyclin B, PLK-1, and both kinases combined as indicated. Phosphorylation was assessed using SDS-PAGE followed by Coomassie (total protein) or ProQ diamond (phosphoprotein) staining. (**E**) Western blot of recombinant BUB-1190-628 Cdk1/Cyclin B kinase assays incubated with either α-BUB-1 or α-phT527 BUB-1 antibodies. To ensure specificity, α-phT527 BUB-1 antibody was incubated alone or together with an unphosphorylated or phT527 BUB-1 STP motif peptide, respectively. (**F**) (top) Schematic of BUB-1190-628(T527A) (bottom) kinase assay of recombinant BUB-1190-628 and BUB-1190-628(T527A) with Cdk1/Cyclin B and PLK-1 combined. Proteins were incubated with the kinases for the indicated time points before phosphorylation was assessed by SDS-PAGE and staining with either Coomassie (total protein) or ProQ diamond (phosphoprotein).

The online version of this article includes the following source data and figure supplement(s) for figure 2:

*Figure 2 continued on next page*

*Figure 2 continued*

**Source data 1.** Gel/blot images for data presented in *Figure 2*.

**Figure supplement 1.** BUB-1 phosphorylation by Cdk1 and PLK-1 in vitro.

**Figure supplement 1—source data 1.** Gel scan images for data presented in *Figure 2—figure supplement 1*.

To further confirm that the STP motif of BUB-1 can interact with PLK1$^{PBD}$ in a phT527-dependent manner, FITC-labelled peptides containing the BUB-1 STP motif were incubated with either wild type MBP-PLK1$^{PBD}$ or a mutant that cannot interact with phSTP motifs ('MBP-PLK1$^{PBD}$(H538A/K540M)') (*Elia et al., 2003b*). The phT527 peptide bound to MBP-PLK1$^{PBD}$ with a $K_D$ of 55 ± 3 nM but did not bind to MBP-PLK1$^{PBD}$(H538A/K540M), while the unphosphorylated peptide did not interact with MBP-PLK1$^{PBD}$ at the concentrations tested (*Figure 3D*).

Altogether, these data indicate that *C. elegans* BUB-1 can directly bind to PLK1$^{PBD}$ in vitro in a phospho-dependent manner via a newly characterised STP motif.

## BUB-1 directly recruits PLK-1 to the midbivalent in vivo

We then sought to determine whether the STP motif in BUB-1 is responsible for PLK-1 recruitment in vivo. Using CRISPR-Cas9, we generated the T527A mutation in the endogenous *bub-1* locus (*bub-1$^{T527A}$*). *bub-1$^{T527A}$* mutant worms showed significant embryonic and larval lethality so we generated a balanced strain in which the *bub-1$^{T527A}$* allele was maintained as a heterozygote. Homozygous *bub-1$^{T527A}$* worms from heterozygous parents develop to adulthood and produce oocytes that go through meiosis, which allowed us to study the role of the STP motif in BUB-1 during meiosis. PLK-1 was absent from the midbivalent in *bub-1(T527A)* oocytes, reminiscent of the *bub-1(RNAi)* phenotype (*Figure 4A*, blue arrowheads, and B). Importantly, BUB-1 localisation to the midbivalent and kinetochore was maintained in the *bub-1$^{T527A}$* strain (*Figure 4C*), indicating that BUB-1 directly interacts with PLK-1 via this STP motif in vivo to recruit PLK-1 to the midbivalent. To assess the impact of BUB-1 mediated PLK-1 recruitment during meiosis I, we crossed the *bub-1$^{T527A}$* allele with a strain expressing GFP-tagged tubulin and mCherry-tagged histone and analysed chromosome alignment, segregation, and polar body extrusion defects (see 'Methods'). *bub-1$^{T527A}$* mutant oocytes displayed chromosome alignment defects in ~62% of the oocytes, with 32% of oocytes showing severe alignment defects (*Figure 4D and E*). Additionally, ~1/4 of bub-1$^{T527A}$ oocytes showed mild anaphase defects (*Figure 4E*). Despite these defects, more than 90% of *bub-1$^{T527A}$* oocytes show visible separation of two chromosome masses and polar body extrusion occurred normally at the end of meiosis I (*Figure 4E*).

Overall, our results show that an STP motif in BUB-1 directly recruits PLK-1 to the midbivalent during meiosis I, and this interaction is primarily important for chromosome alignment.

## CENP-C$^{HCP-4}$ recruits PLK-1 to meiotic chromosome arms

PLK-1 localisation to chromosome arms remained unchanged when BUB-1 was depleted (*Figure 2A and B*) or the STP motif was mutated (*Figure 4A and B*), indicating that a different pathway is required to recruit this population of PLK-1. We therefore sought to identify the mechanism of PLK-1 recruitment to the chromosome arms. In mammalian mitosis, PLK1 recruitment to the kinetochore is mediated by BUB-1 and CCAN component CENP-U (*Elowe et al., 2007*; *Kang et al., 2006*; *Kang et al., 2011*; *Qi et al., 2006*; *Singh et al., 2021*). Interestingly, the CCAN appears to be largely absent in *C. elegans* (*Maddox et al., 2012*), and kinetochore assembly depends on the CENP-C orthologue, HCP-4 (hereafter CENP-C$^{HCP-4}$) during mitosis (*Oegema et al., 2001*) and on CENP-C$^{HCP-4}$ and the nucleoporin ELYS$^{MEL-28}$ during meiosis (*Hattersley et al., 2022*). CENP-C$^{HCP-4}$ localises to chromosomes throughout meiosis I (*Figure 5A*; *Hattersley et al., 2022*; *Monen et al., 2005*). Like its mammalian counterpart, CENP-C$^{HCP-4}$ is predicted to be mostly disordered and it contains a putative N-terminal STP motif encompassing amino acids 162–164 that is conserved in nematode species (*Figure 5B*). Although CENP-C$^{HCP-4}$ depletion does not have a major impact on meiosis I (*Hattersley et al., 2022*; *Monen et al., 2005*), RNAi-mediated depletion of CENP-C$^{HCP-4}$ abolished PLK-1 localisation on chromosome arms (*Figure 5C and D*). PLK-1 is still present in the midbivalent (*Figure 5C and D*, blue arrows), which suggests that BUB-1 and CENP-C$^{HCP-4}$ represent independent pathways for PLK-1 targeting. Additionally, CENP-C$^{HCP-4}$ depletion revealed a pool of PLK-1 which is kinetochore-associated (*Figure 5C and D*, yellow arrows). To confirm that PLK-1 localises to the kinetochore, we took advantage of the

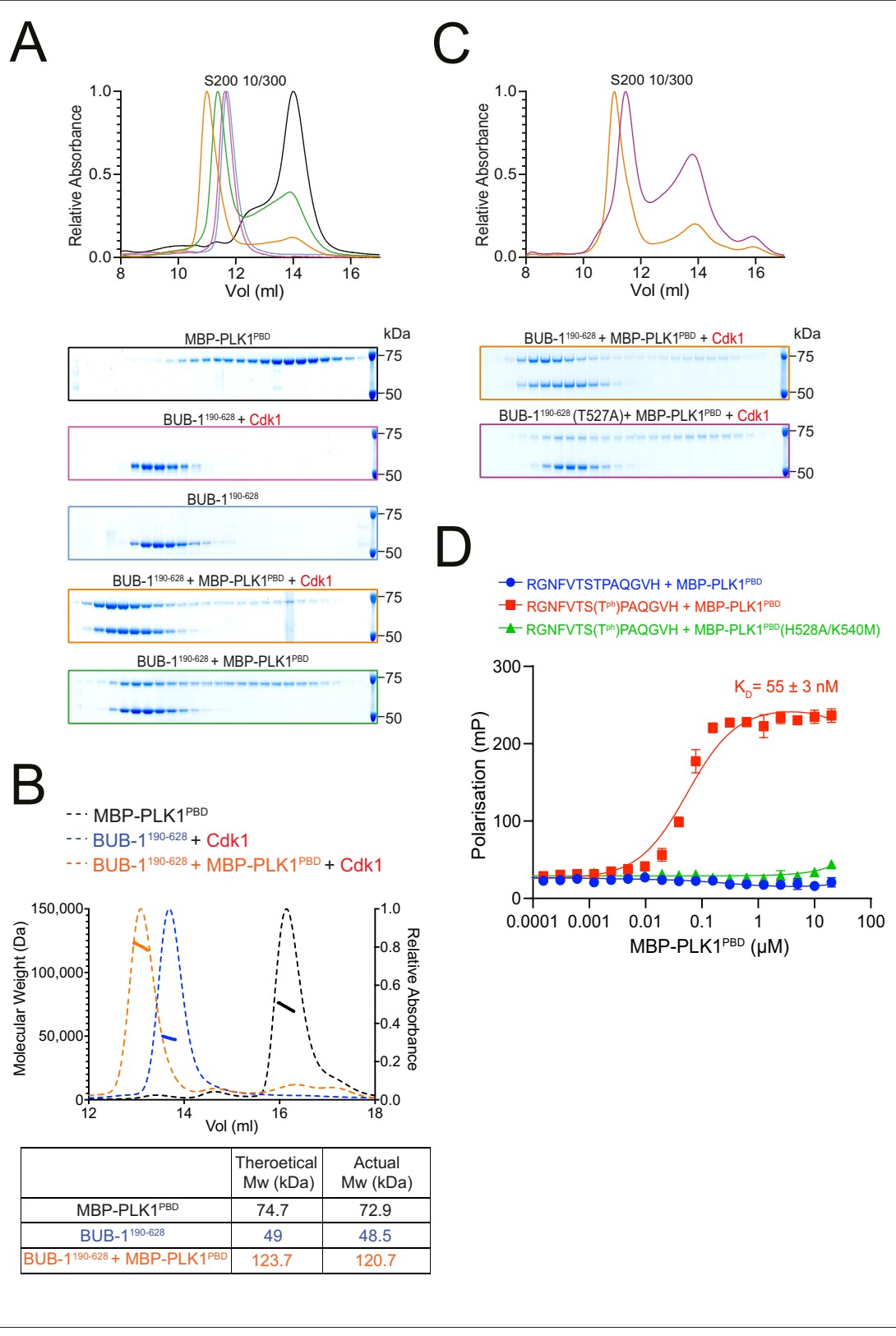

**Figure 3.** BUB-1 directly binds to PLK1 in vitro via a Cdk1-phosphorylated STP motif. (**A**) Elution profile and Coomassie-stained SDS-PAGE gels of representative fractions from the Superdex 200 10/300 SEC column. BUB-1[190-628] was incubated with MBP-PLK1[PBD] at equimolar concentrations before separation by size-exclusion chromatography (SEC). Binding was conducted with unphosphorylated or Cdk1/Cyclin B phosphorylated BUB-1[190-628] as indicated. The colours of the gel borders correspond to the colours of the respective elution profile. (**B**) SEC coupled with multi-angle light scattering

*Figure 3 continued on next page*

*Figure 3 continued*

(SEC-MALS) analysis of BUB-1$^{190-628}$ and MBP-PLK1$^{PBD}$ binding indicates a 1:1 complex stoichiometry. Relative absorbance (dotted lines) and molecular weight (solid lines) are colour-coded to match the corresponding proteins. (C) Elution profile and Coomassie-stained SDS-PAGE gels of representative fractions from the Superdex 200 10/300 SEC column. Wild type or T527A mutant BUB-1$^{190-628}$ was phosphorylated by Cdk1/Cyclin B before incubation with an equimolar concentration of MBP-PLK1$^{PBD}$, binding was assessed by SEC. The colours of the gel borders correspond to the colours of the respective elution profile. (D) FITC-labelled peptides containing the BUB-1 STP motif were incubated with increasing concentrations of MBP-PLK1$^{PBD}$ or MBP-PLK-1$^{PBD}$(H538A/K540M) and binding analysed by fluorescence polarisation. 'T$^{Ph}$' denotes phosphorylated threonine. The colours of the data points correspond to the colours of the experimental conditions as indicated. Data represents mean of 3 independent experiments with error bars/uncertainties denoting the standard deviation between these.

The online version of this article includes the following source data for figure 3:

**Source data 1.** Full gel images for data presented in *Figure 3*.

---

fact that kinetochore proteins in *C. elegans* oocytes are also present in filaments referred to as 'linear elements', which are spindle- and cortex-associated (*Monen et al., 2005*). PLK-1 perfectly co-localises with KNL-3, part of the Mis12 kinetochore complex in spindle- and cortex-associated linear elements (*Figure 5—figure supplement 1*). Since the above data indicated that CENP-C$^{HCP-4}$ is involved in PLK-1 recruitment to chromosome arms, we sought to determine whether this involved a direct interaction via the putative STP motif identified in sequence alignments.

## CENP-C$^{HCP-4}$ interacts directly with PLK-1 through a Cdk1-dependent STP motif

To investigate whether CENP-C$^{HCP-4}$ directly interacts with PLK-1, we purified a recombinant N-terminal fragment of CENP-C$^{HCP-4}$ ('CENP-C$^{HCP-4(1-214)}$', *Figure 5E*). As the putative CENP-C$^{HCP-4}$ STP motif is also a proline-directed kinase consensus site, we assessed Cdk1 phosphorylation of the recombinant fragment. Cdk1/Cyclin B kinase assays showed that CENP-C$^{HCP-4(1-214)}$ can be phosphorylated by Cdk1, and mutation of the putative STP motif threonine to alanine (T163A) reduced the phosphorylation of the fragment (*Figure 5E and F*). Furthermore, Cdk1 specifically phosphorylated T163 within the putative STP motif (*Figure 5G*).

To determine whether CENP-C$^{HCP-4(1-214)}$ can directly bind to MBP-PLK1$^{PBD}$, we used SEC. CENP-C$^{HCP-4(1-214)}$ formed a stable complex with MBP-PLK1$^{PBD}$ which was dependent on Cdk1-phosphorylation (*Figure 6A*). SEC-MALS indicated the stoichiometry of this interaction was 1:1 (*Figure 6B*). We then tested whether phosphorylation of the STP motif is required for the interaction with MBP-PLK1$^{PBD}$. Mutation of the putative STP motif (T163A) largely prevented the Cdk1-mediated interaction with MBP-PLK1$^{PBD}$, indicating phosphorylation of the STP motif at T163 is required for the interaction (*Figure 6C*).

To further confirm this STP motif binds to MBP-PLK1$^{PBD}$ in a phT163-dependent manner, we conducted fluorescence polarisation assays with FITC-labelled CENP-C$^{HCP-4}$ STP motif peptides (*Figure 6D*). The phT163 peptide bound to MBP-PLK1$^{PBD}$ with a $K_D$ of 104 ± 14 nM but did not interact with MBP-PLK1$^{PBD}$(H538A/K540M), while the unphosphorylated peptide did not bind to MBP-PLK1$^{PBD}$ at the concentrations tested (*Figure 6D*). Collectively, these data indicate that the putative STP motif in CENP-C$^{HCP-4}$ binds to the PBDs of PLK-1 in a Cdk1 phosphorylation-dependent manner, which led us to assess the importance of this STP motif in vivo.

## CENP-C$^{HCP-4}$ recruits PLK-1 to chromosome arms in vivo through an STP motif

The T163A mutation in CENP-C$^{HCP-4}$ was generated in the endogenous *hcp-4* locus (*hcp-4$^{T163A}$*) which, unlike *bub-1$^{T527A}$*, did not affect viability. When PLK-1::sfGFP was monitored in dissected oocytes, *hcp-4$^{T163A}$* recapitulated the full CENP-C$^{HCP-4}$ depletion with PLK-1 localising to the midbivalent and kinetochore but largely absent from chromosome arms (*Figure 7A and B*). This indicates that PLK-1 is targeted to chromosome arms directly through the phospho-dependent STP motif in CENP-C$^{HCP-4}$. Importantly, CENP-C$^{HCP-4}$(T163A) displays an indistinguishable localisation from wild type CENP-C$^{HCP-4}$ (*Figure 7C*) and the other PLK-1 receptor, BUB-1, also localises normally in the *hcp-4$^{T163A}$* strain (*Figure 7D*).

These data indicate that CENP-C$^{HCP-4}$ directly recruits PLK-1 to the chromosome arms during meiosis I via a newly characterised N-terminal STP motif.

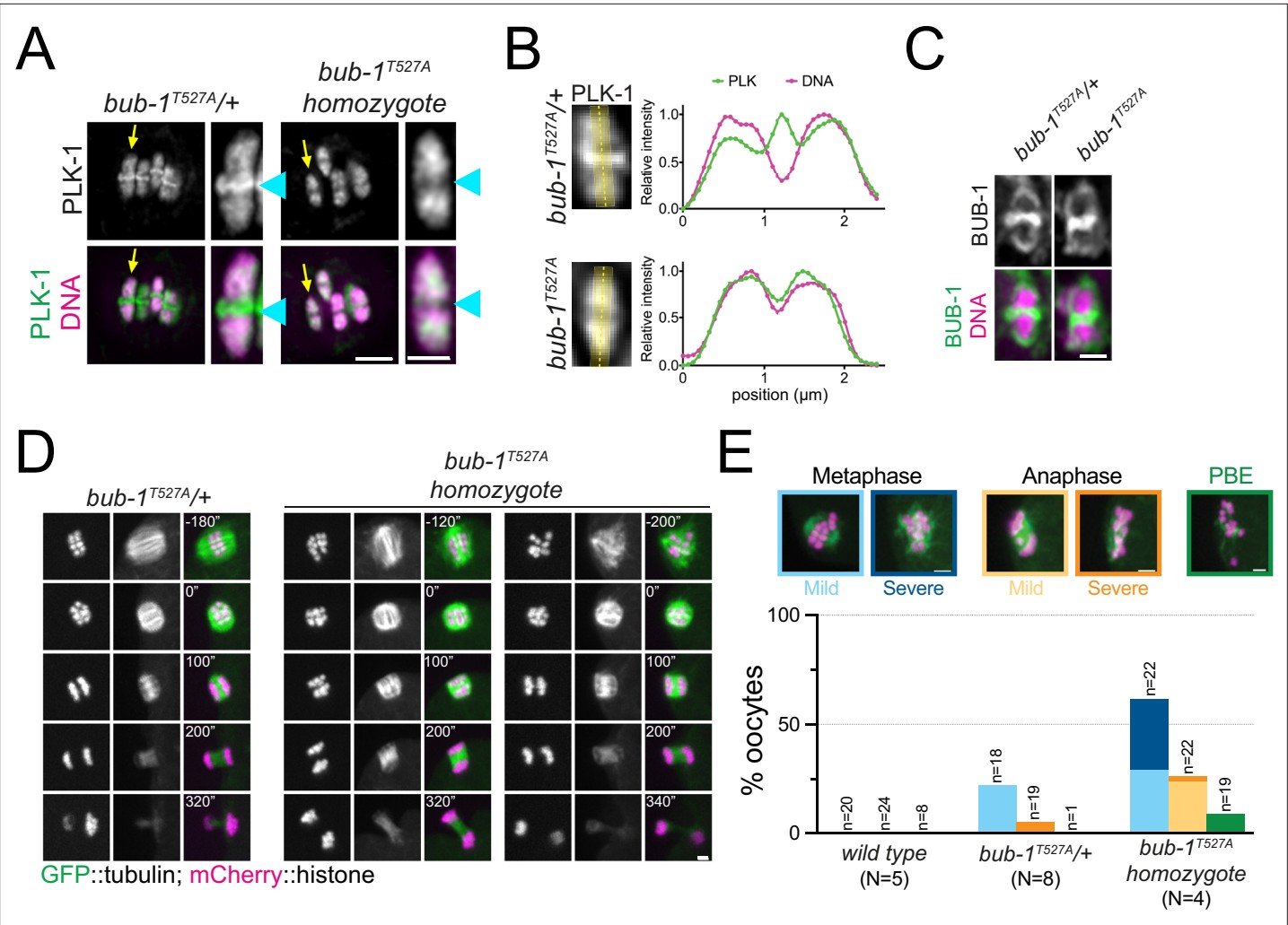

**Figure 4.** The BUB-1 STP motif is required for PLK-1 targeting and chromosome alignment. (**A**) Fixed oocytes were stained with a PLK-1-specific antibody (green), and *bub-1^T527A* heterozygote and homozygote oocytes were compared. DNA is shown in magenta. The yellow arrow points to the midbivalent magnified on the right in each case, and the blue arrowhead points to the midbivalent. Scale bars, 2 µm (left) and 1 µm (right). (**B**) Line profile analysis of PLK-1 localisation in fixed oocytes during early metaphase I in *bub-1^T527A* heterozygote ('*bub-1^T527A/+*') and homozygote ('*bub-1^T527A*') oocytes, as indicated by the yellow lines. Background signal was subtracted, and maximum signal for each channel was set to 1. (**C**) Fixed oocytes were stained with a BUB-1-specific antibody (green), and *bub-1^T527A* heterozygote and homozygote oocytes were compared. DNA is shown in magenta. Scale bar, 1 µm. (**D**) *bub-1^T527A* heterozygote ('*bub-1^T527A/+*') and homozygote ('*bub-1^T527A*') oocytes expressing GFP-tagged tubulin and mCherry-tagged histone were filmed during meiosis I. Two homozygote ('*bub-1^T527A*') oocytes are shown to depict the difference in severity of the alignment defect. Scale bar, 2 µm. (**E**) Meiotic defects (as described in the 'Methods' section) were assessed in wild type, *bub-1^T527A* heterozygote ('*bub-1^T527A/+*') and homozygote oocytes. Representative images of the different phenotypes analysed are presented on top (scale bars, 2 µm). 'N' represents the number of experiments, and 'n' denotes the number of oocytes analysed.

The online version of this article includes the following source data for figure 4:

**Source data 1.** Data utilised to generate the graph in *Figure 4*.

## Dual PLK-1 recruitment by BUB-1 and CENP-C is essential for meiosis I

Our results so far indicate that PLK-1 recruitment to the midbivalent during meiosis I is mediated by direct interaction with BUB-1, while PLK-1 localisation to the chromosome arms requires direct interaction with CENP-C^HCP-4. Co-depletion of BUB-1 and CENP-C^HCP-4 by RNAi led to a drastic reduction in total PLK-1 signal (15% median intensity of wild type oocytes; *Figure 8A*, *Figure 8—figure supplement 1*). This confirmed that the BUB-1 and CENP-C^HCP-4 pathways are the primary recruiters of PLK-1 to the chromosomes during oocyte meiosis. Additionally, the kinetochore population of PLK-1 was lost when BUB-1 and HCP-4 were co-depleted, indicating that BUB-1 is responsible for recruiting

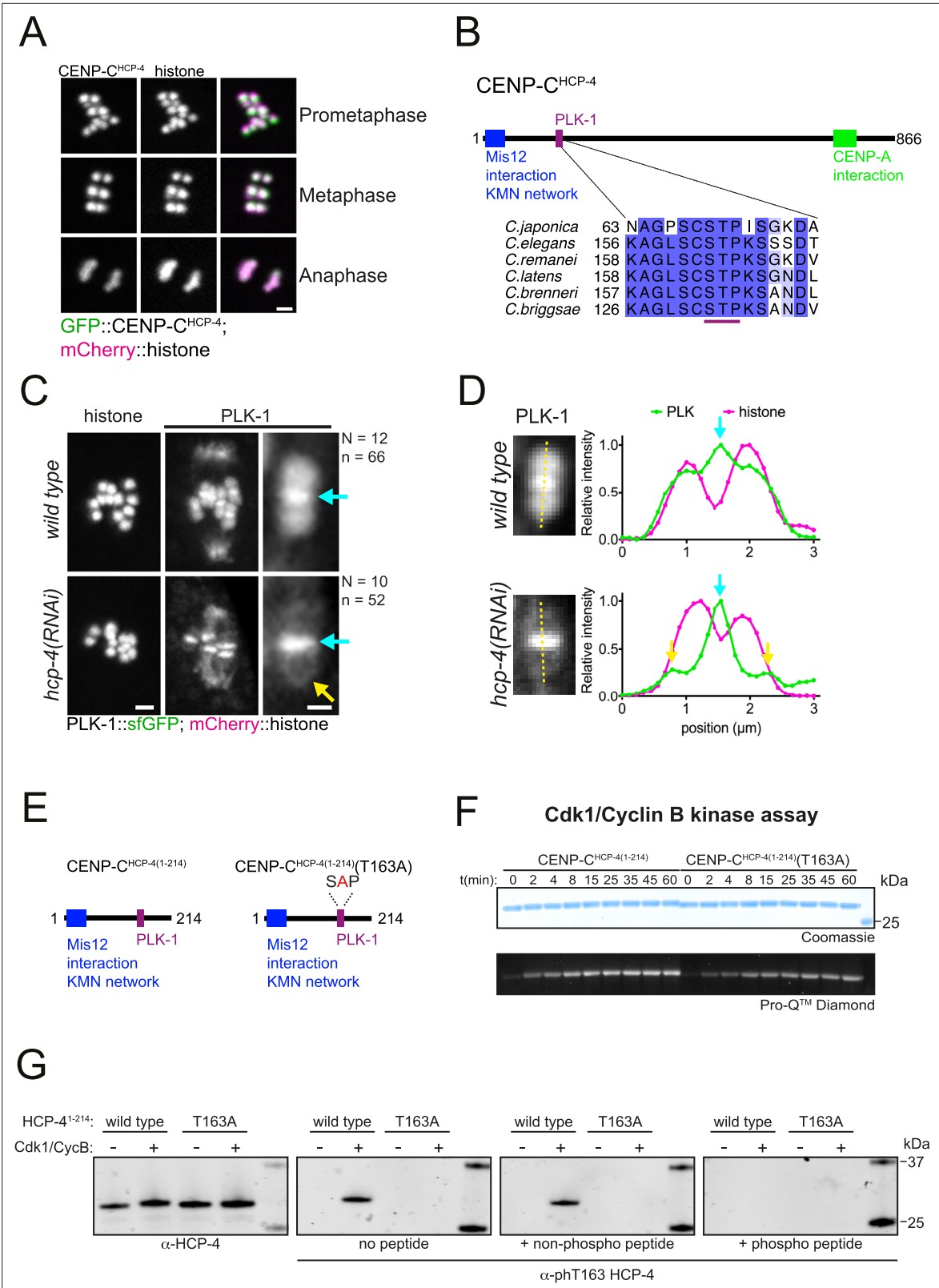

**Figure 5.** CENP-C[HCP-4] contains a putative STP motif and is required for chromosomal PLK-1 targeting. (**A**) In situ GFP-tagged CENP-C[HCP-4] was followed through meiosis I in live, dissected oocytes. Scale bar, 2 μm. (**B**) Schematic representation of the *C. elegans* CENP-C[HCP-4] (top) and sequence alignment of the putative STP motif in nematode species (bottom). (**C**) Control ('*wild type*') and CENP-C[HCP-4]-depleted ['*hcp-4(RNAi)*'] oocytes expressing sfGFP-tagged PLK-1 (and mCherry-tagged histone) were dissected and recorded throughout meiosis I. Prometaphase I is shown (before spindle rotation/

*Figure 5 continued on next page*

*Figure 5 continued*

shortening). On the right, the cyan arrows point to the midbivalent, and the yellow arrow points to the kinetochore. 'N' and 'n' denote the number of oocytes and bivalents analysed, respectively. Left scale bar, 2 μm. Right scale bar, 1 μm. (**D**) Line profile analysis of PLK-1::sfGFP during early metaphase I in *wild type* and *hcp-4(RNAi)* oocytes, as indicated by the yellow lines. Background signal was subtracted, and maximum signal for each channel was set to 1. The blue arrows point to the midbivalent, and the yellow arrow points to the kinetochore. (**E**) Schematic of recombinant CENP-C$^{HCP-4(1-214)}$ proteins. (**F**) Kinase assay of recombinant CENP-C$^{HCP-4(1-214)}$ wild type and T163A proteins with Cdk1:Cyclin B. Reactions were analysed by SDS-PAGE followed by ProQ diamond (phosphoprotein) or Coomassie (total protein) staining. (**G**) Western blot of recombinant CENP-C$^{HCP-4(1-214)}$ Cdk1/Cyclin B kinase assays incubated with either α-HCP-4 or α-phT163 HCP-4 antibodies. To ensure specificity, α-phT163 HCP-4 antibody was incubated alone or together with an unphosphorylated or phT163 HCP-4 STP motif peptide, respectively.

The online version of this article includes the following source data and figure supplement(s) for figure 5:

**Source data 1.** Gel/blot images for data presented in *Figure 5*.

**Figure supplement 1.** PLK-1 co-localises with the kinetochore component KNL-3 in non-chromosomal kinetochore assemblies.

PLK-1 to the kinetochore as well as the midbivalent (*Figure 8A*). RNAi of BUB-1 causes significant defects in spindle stability, chromosome alignment, and chromosome segregation during meiosis (*Dumont et al., 2010*; *Pelisch et al., 2019*). However, co-depletion of BUB-1 and CENP-C$^{HCP-4}$ exacerbated the chromosome alignment and segregation errors and resulted in a significantly higher proportion of polar body extrusion failures (*Figure 8B and C*). Therefore, while CENP-C$^{HCP-4}$ depletion does not have noticeable defects on its own, it enhances the BUB-1 depletion phenotype. To focus more specifically on the effects of direct PLK-1 recruitment, we depleted CENP-C$^{HCP-4}$ in *bub-1$^{T527A}$* mutant oocytes. This largely abolished PLK-1 localisation in the same manner as co-depletion of BUB-1 and CENP-C$^{HCP-4}$ (*Figure 8D*) as well as displaying defects in chromosome alignment, segregation, and polar body extrusion of a greater severity than *hcp-4(RNAi)* or *bub-1$^{T527A}$* alone (*Figure 8E and F*). Similar results were obtained when performing the complementary experiment using *hcp-4$^{T163A}$/bub-1(RNAi)* oocytes (*Figure 8—figure supplement 2*).

To determine the specific, combined role of the STP motifs in BUB-1 and CENP-C$^{HCP-4}$, we used a gene replacement strategy whereby RNAi-resistant wild type or T527A BUB-1 transgenes ('BUB-1$^{wt}$' and 'BUB-1$^{T527A}$', respectively) were expressed in the presence of *bub-1(RNAi)* to deplete the endogenous protein (*Figure 9—figure supplement 1A and B*). We then used this strategy in wild type or *hcp-4$^{T163A}$* mutant worms ('*hcp-4$^{wt}$*' and '*hcp-4$^{T163A}$*', respectively) to assess the effects of the double STP mutant. PLK-1 localised normally to the chromosomes and midbivalent in BUB-1$^{wt}$, but was completely absent from the midbivalent in BUB-1$^{T527A}$ (*Figure 9A*, *Figure 9—figure supplement 2A*, cyan arrows). In the *hcp-4$^{T163A}$* mutant PLK-1 was absent from chromosome arms when BUB-1$^{wt}$ was expressed (*Figure 9A*; *Figure 9—figure supplement 2A*, yellow arrows), and it was largely absent from all chromosome-associated domains in the presence of BUB-1$^{T527A}$ (*Figure 9A*, *Figure 9—figure supplement 2A*).

We then used the same approach to assess the combined effect of the double mutant *bub-1$^{T527A}$/hcp-4$^{T163A}$* on chromosome alignment and segregation, imaging oocytes expressing GFP-tagged tubulin and mCherry-tagged histone. While mild defects were observed in the bub-1$^{T527A}$ mutant (and to a lesser extent in the hcp-4$^{T163A}$ mutant), the double bub-1$^{T527A}$/hcp-4$^{T163A}$ mutant displayed severe meiotic defects, with 95% of oocytes displaying severe metaphase defects, ~70% displaying severe anaphase defects, and ~73% of oocytes failing to achieve PBE (*Figure 9B and C*, *Figure 9—figure supplement 2*).

These data indicate that recruitment of PLK-1 to the midbivalent and kinetochore by BUB-1 appears to be primarily responsible for the chromosomal roles of PLK-1 during meiosis I, as disruption of this pathway leads to meiotic defects while perturbing CENP-C$^{HCP-4}$ recruitment of PLK-1 on its own does not. However, the fact that disruption of both BUB-1 and CENP-C$^{HCP-4}$ recruitment pathways enhances the severity of the resulting meiotic defects indicates that the CENP-C$^{HPC-4}$ pathway does still play an active part in the roles of PLK-1 during meiosis I.

## Discussion

In this article, we describe specific, post-NEBD roles played by PLK-1 during oocyte meiosis. PLK-1 is important for spindle assembly and/or stability, chromosome alignment and segregation, and polar body extrusion during meiosis I. We found that PLK-1 localises to spindle poles and chromosomes

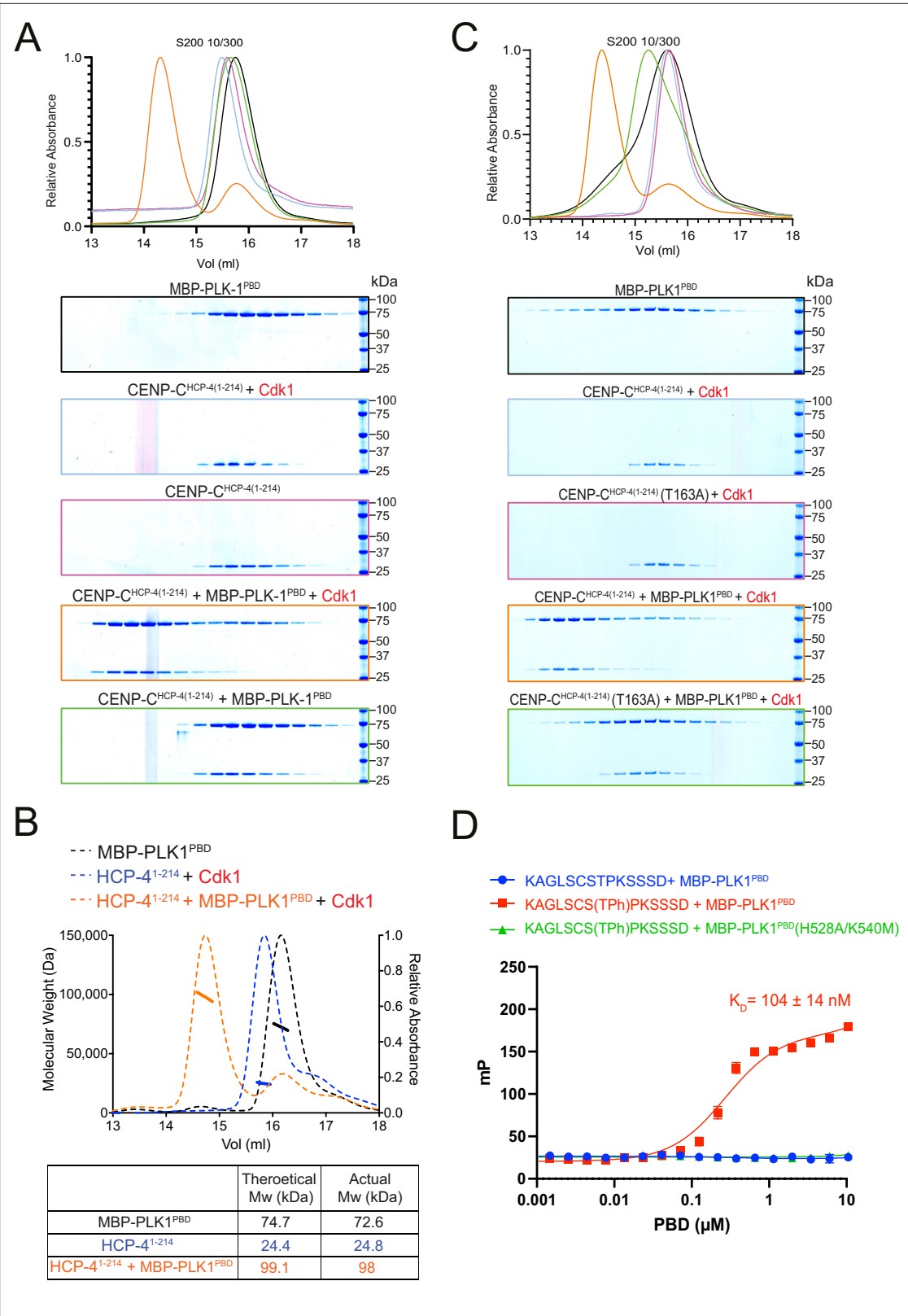

**Figure 6.** CENP-C[HCP-4] interacts with PLK-1 in vitro in a Cdk1-dependent manner. (**A**) Elution profile and Coomassie-stained SDS-PAGE gels of representative fractions from the Superdex 200 10/300 size-exclusion chromatography (SEC) column. CENPC-C[HCP-4(1-214)] was incubated with MBP-PLK1[PBD] at equimolar concentrations before being analysed by SEC. Binding was conducted with unphosphorylated or Cdk1:Cyclin B phosphorylated CENP-C[HCP-4(1-214)] as indicated. The colours of the gel borders correspond to the colours of the respective elution profile. (**B**) Size-exclusion chromatography

*Figure 6 continued on next page*

*Figure 6 continued*

coupled with multi-angle light scattering (SEC-MALS) analysis of CENP-C$^{HCP-4(1-214)}$ and MBP-PLK1$^{PBD}$ binding indicates a 1:1 complex stoichiometry. Relative absorbance (dotted lines) and molecular weight (solid lines) are colour-coded to match the corresponding proteins. (**C**) Elution profile and Coomassie-stained SDS-PAGE gels of representative fractions from the Superdex 200 10/300 SEC column. Wild type or T163A mutant CENP-C$^{HCP-4(1-214)}$ was phosphorylated by Cdk1/Cyclin B before incubation with an equimolar concentration of MBP-PLK1$^{PBD}$. Binding was then assessed by SEC. The colours of the gel borders correspond to the colours of the respective elution profile. (**D**) FITC-labelled peptides containing the HCP-4 STP motif were incubated with increasing concentrations of MBP-PLK1$^{PBD}$ or MBP-PLK-1$^{PBD}$(H538A/K540M) and binding analysed by fluorescence polarisation. Unphosphorylated versus T163-phosphorylated (T$^{Ph}$) peptides were compared. The colours of the data points correspond to the colours of the experimental conditions as indicated. Data represents mean of 3 independent experiments with error bars/uncertainties denoting the standard deviation between these.

The online version of this article includes the following source data for figure 6:

**Source data 1.** Full gel images for data presented in *Figure 6*.

during metaphase I before localising to the chromosomes and central spindle in anaphase. Furthermore, we characterised the mechanisms of chromosomal PLK-1 targeting during oocyte meiosis, which rely on the centromere-associated protein CENP-C$^{HCP-4}$ and the spindle assembly checkpoint kinase BUB-1. While CENP-C$^{HCP-4}$ targets PLK-1 to chromosome arms, BUB-1 directs PLK-1 to the midbivalent and kinetochores. In both cases, interaction with PLK-1 relies on phosphorylated STP motifs within predicted disordered regions. While we have not confirmed that these sites are phosphorylated by Cdk1 in vivo, several lines of evidence indicate this is likely the case: (1) both motifs have Pro at position 3, indicative of potential proline-directed kinase substrates; (2) Cdk1 is the most prominent proline-directed kinase in cell division and known to phosphorylate STP motifs, including that of mammalian BUB1 (*Qi et al., 2006*); (3) both sites were phosphorylated by Cdk1 in vitro; and (4) fluorescence polarisation experiments indicate that phosphorylation of the Thr residues within the STP motifs is essential for the interaction with the PBD, clearly displaying why the alanine mutants have such a drastic effect on PLK-1 localisation in vivo. We found that mutating the STP motif in BUB-1 results primarily in chromosome alignment defects. While disrupting the CENP-C$^{HCP-4}$-mediated localisation of PLK-1 to the chromosome arms does not have a significant phenotypic defect on its own, it does enhance the meiotic defects observed when the BUB-1-dependent kinetochore and midbivalent populations are disrupted. This suggests that while BUB-1 recruitment of PLK-1 may mediate the most important functions of PLK-1 during meiosis I, CENP-C$^{HCP-4}$ recruitment of PLK-1 to the chromosome arms also plays an active role in meiosis.

## Expanding on the meiotic roles of PLK-1

By temporally inhibiting an analogue-sensitive PLK-1 mutant during *C. elegans* oocyte meiosis, we have shown that PLK-1 plays a major role in the regulation of the meiotic spindle. At high concentrations of analogue, this resulted in the complete lack of spindle bipolarity, and even at low concentrations the majority of oocytes (62%) imaged lacked an apparent bipolar spindle. While this clearly displays a key role of PLK-1 during meiosis is in the regulation of the meiotic spindle, we cannot distinguish more specific mechanisms using our experimental techniques. As a result, we have characterised the phenotype as a lack of spindle stability throughout this article, but it should be noted that we cannot distinguish whether these effects are on the spindle assembly process itself or on the maintenance of an assembled bipolar spindle. Potential mechanisms of meiotic spindle regulation by PLK-1 include the microtubule depolymerase KLP-7, the *C. elegans* ortholog of the MCAK/Kinesin 13 family. There is some evidence to indicate that PLK-1 may regulate the kinesin 13 microtubule depolymerases in other organisms (*Jang et al., 2009*; *Ritter et al., 2014*; *Sanhaji et al., 2014*; *Sanhaji et al., 2014*; *Shao et al., 2015*; *Zhang et al., 2011*) and KLP-7 localises to the chromosomes, during meiosis I (*Connolly et al., 2015*; *Danlasky et al., 2020*; *Gigant et al., 2017*; *Han et al., 2015*), which overlaps with PLK-1 localisation. Furthermore, disrupting KLP-7 function prevents proper bipolar spindle assembly and results in microtubules protruding out of the meiotic spindle towards the cytoplasm (*Connolly et al., 2015*; *Gigant et al., 2017*), two phenotypes we also see with PLK-1 inhibition.

Aside from the large-scale spindle defects observed when PLK-1 is inhibited, chromosome alignment is still disrupted when the structure of the spindle appears bipolar and largely normal. Since the mechanism of chromosome alignment by acentrosomal meiotic spindles is not as well characterised as the centrosomal mitotic equivalent, speculating on the underpinning mechanisms of this phenotype is

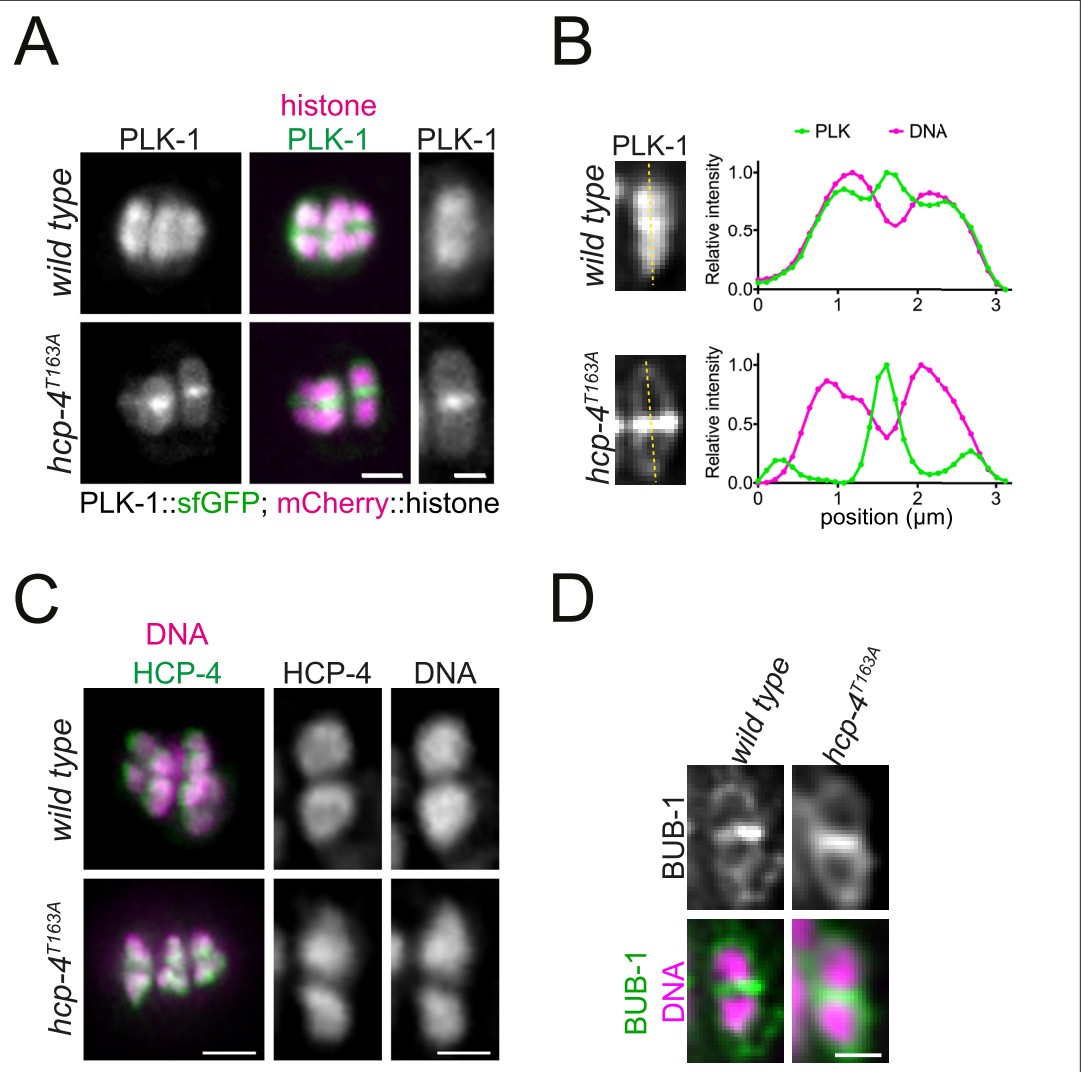

**Figure 7.** CENP-C[HCP-4] directly recruits PLK-1 to chromosome arms. (**A**) Control ('*wild type*') and CENP-C[HCP-4] STP mutant ['*hcp-4[T163A]*'] oocytes expressing sfGFP-tagged PLK-1 (and mCherry-tagged histone) were dissected and recorded throughout meiosis I. Metaphase I is shown. Left scale bar, 2 µm. Right scale bar, 1 µm. (**B**) Line profile analysis of PLK-1::sfGFP during early metaphase I in *wild type* and *hcp-4[T163A]* oocytes, as indicated by the yellow lines. Background signal was subtracted, and maximum signal for each channel was set to 1. (**C**) Fixed oocytes were stained with an HCP-4 specific antibody (green in the merged image). *hcp-4[T163A]* mutant oocytes were compared to wild type. DNA is shown in magenta in the merged panel. Scale bar on the left, 2 µm; scale bar on the right, 1 µm. (**D**) Same as in (**C**) but using a BUB-1-specific antibody to compare BUB-1 localisation in wild type and *hcp-4[T163A]* mutant oocytes. Scale bar, 1 µm.

challenging. However, it should be noted that we cannot exclude the possibility that this phenotype is also a direct result of the dysregulated spindle upon PLK-1 inhibition. Indeed, there is some evidence to suggest that a chromosome-dependent pathway of microtubule formation may be an important aspect of chromosome alignment and segregation (*Conway et al., 2022*; *Heald et al., 1996*; *Kiewisz et al., 2022*).

Despite these specific hypotheses mentioned above, there will be many different proteins and pathways impacted by PLK-1 phosphorylation throughout meiosis I that will ultimately contribute to the severe defects we observe upon PLK-1 inhibition. While a focussed investigation into specific hypotheses would no doubt yield important results, an unbiased approach to identify the relevant PLK-1 substrates during meiosis would be particularly useful for investigation of the key effects of PLK-1 during meiosis. There are obvious technical challenges to overcome before this can be achieved, not

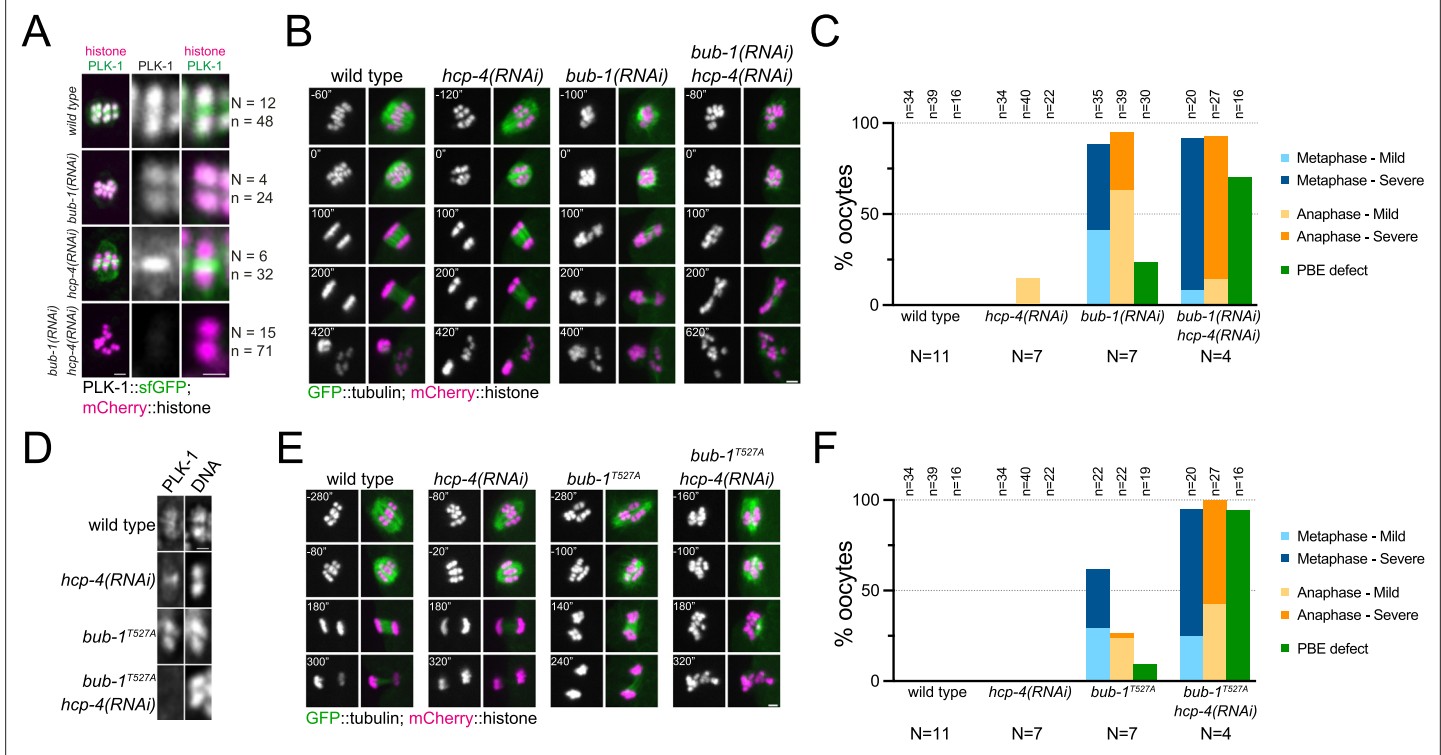

**Figure 8.** Combined disruption of BUB-1- and CENP-C^HCP-4-dependent PLK-1 recruitment leads to severe meiotic defects. (**A**) Control ('*wild type*'), BUB-1-depleted ['*bub-1(RNAi)*'], and CENP-C^HCP-4-depleted ['*hcp-4(RNAi)*'] oocytes expressing sfGFP-tagged PLK-1 (and mCherry-tagged histone) were dissected and recorded throughout meiosis I. Prometaphase/metaphase I is shown (before spindle rotation/shortening). 'N' and 'n' denote the number of oocytes and bivalents analysed, respectively. Left scale bar, 2 μm. Right scale bar, 1 μm. (**B**) Control ('*wild type*'), BUB-1-depleted ['*bub-1(RNAi)*'], and CENP-C^HCP-4-depleted ['*hcp-4(RNAi)*'] oocytes expressing GFP-tagged tubulin (and mCherry-tagged histone) were dissected and recorded throughout meiosis I. Scale bar, 2 μm. (**C**) Meiotic defects (as described in the 'Methods' section) were assessed in wild type, *bub-1(RNAi)*, and *hcp-4(RNAi)* oocytes. 'N' and 'n' denote the number of oocytes and bivalents analysed, respectively. (**D**) Fixed wild type, *hcp-4(RNAi)*, *bub-1^T527A*, and *bub-1^T527A+hcp-4(RNAi)* oocytes were stained with a PLK-1-specific antibody. Scale bar, 1 μm. (**E**) Control ('*wild type*'), BUB-1^T527A ['*bub-1^T527A*'], and CENP-C^HCP-4-depleted ['*hcp-4(RNAi)*'] oocytes expressing GFP-tagged tubulin (and mCherry-tagged histone) were dissected and recorded throughout meiosis I. Scale bar, 2 μm. (**F**) Meiotic defects (as described in the 'Methods' section) were assessed in wild type, *bub-1^T527A*, *hcp-4(RNAi)*, and *bub-1^T527A+hcp-4(RNAi)* oocytes. 'N' and 'n' denote the number of oocytes and bivalents analysed, respectively.

The online version of this article includes the following source data and figure supplement(s) for figure 8:

**Source data 1.** Data utilised to generate the graph in *Figure 8C*.

**Source data 2.** Data utilised to generate the graph in *Figure 8F*.

**Figure supplement 1.** PLK-1 intensity measurements.

**Figure supplement 1—source data 1.** Data utilised to generate the graph in *Figure 8—figure supplement 1*, along with the statistical analysis.

**Figure supplement 2.** Combined disruption of BUB-1- and CENP-C^HCP-4-dependent PLK-1 recruitment leads to severe meiotic defects.

**Figure supplement 2—source data 1.** Data utilised to generate the graph in *Figure 8—figure supplement 2*.

least of which would be isolating a large enough sample of meiotic oocytes to perform robust quantitative proteomics. The work in this article undertaken to identify the mechanisms of PLK-1 targeting during oocyte meiosis I will be instrumental for a later characterisation of the localisation and meiotic stage-specific analysis of PLK-1 substrates.

## Comparison with dual recruitment in mammals (CENP-U vs CENP-C)

During mammalian mitosis, PLK1 is recruited to kinetochores through BUB1 and CENP-U, relying on self-priming in addition to Cdk1-mediated priming (*Kang et al., 2006*; *Kang et al., 2011*; *Qi et al., 2006*; *Singh et al., 2021*). Our results suggest that Cdk1-mediated priming is the primary mechanism for PLK-1 recruitment in both BUB-1- and CENP-C-dependent branches. Additionally, we noted the presence of a putative B56 short linear motif (LxxIxE) 38 aa downstream of the STP motif in *C. elegans*

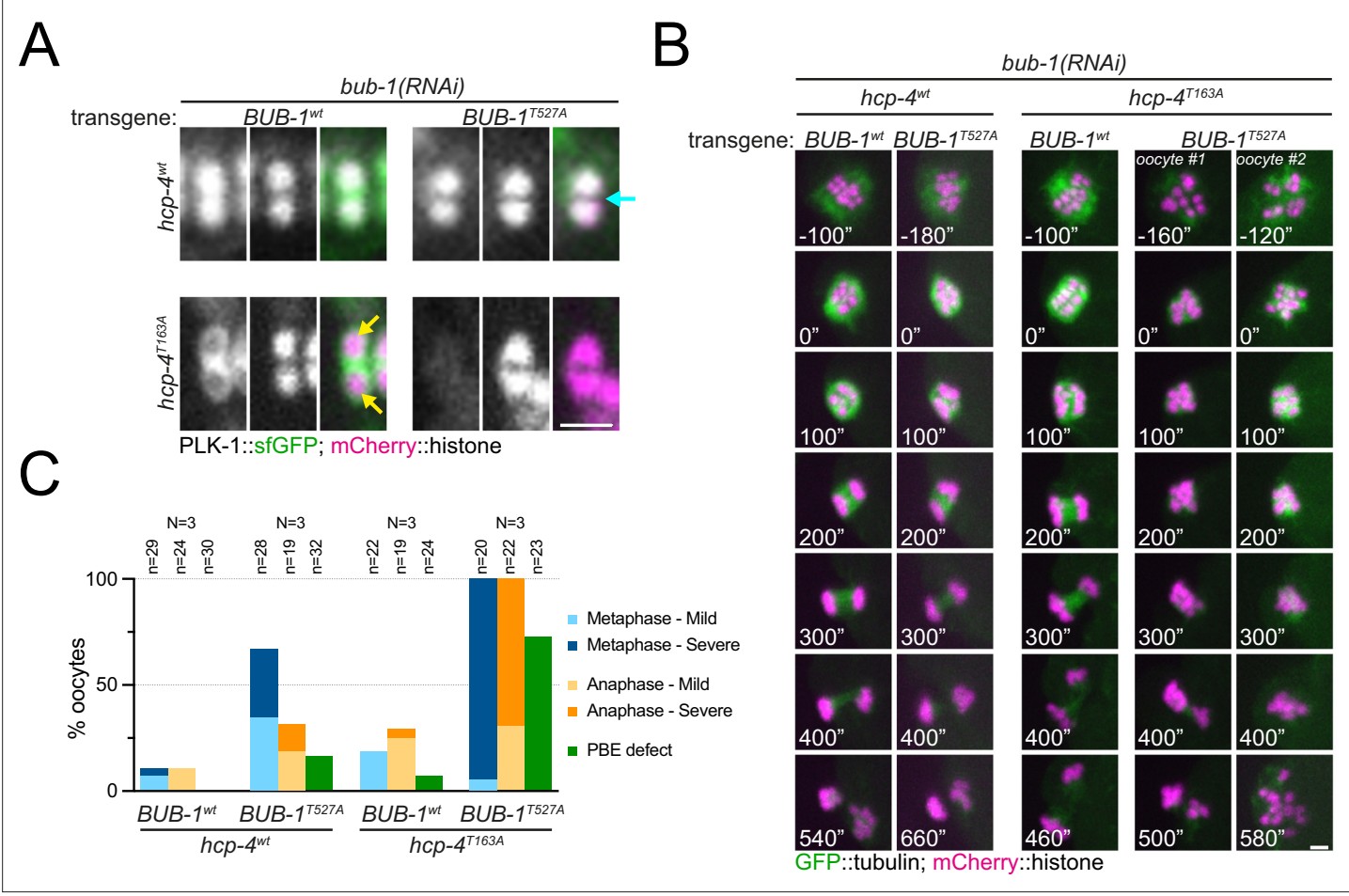

**Figure 9.** Disruption of BUB-1- and CENP-C$^{HCP-4}$ STP motifs leads to severe meiotic defects. (**A**) Wild type BUB-1 ('BUB-1$^{wt}$') and T527A BUB-1 ('BUB-1$^{T527A}$') were expressed from transgenes in the absence of endogenous BUB-1, in wild type ('$hcp-4^{wt}$') or HCP-4 T163A ('$hcp-4^{T163A}$') worms. Oocytes expressing sfGFP-tagged PLK-1 (and mCherry-tagged histone) were dissected and recorded throughout meiosis I. Images from oocytes in prometaphase I are shown. Cyan arrow points to the midbivalent, and yellow arrows point to chromosome arms. Scale bar, 2 μm. (**B**) Oocytes from the same conditions as in (**A**) but expressing GFP-tagged tubulin (and mCherry-tagged histone) were dissected and recorded throughout meiosis I. Scale bar, 2 μm. In the case of the double mutant $hcp-4^{T163A}/bub-1^{T527A}$, two oocytes are shown, one that achieved (defective) PBE ('oocyte #1') and another in which PBE failed ('oocyte #2'). (**C**) Meiotic defects were assessed and are shown as % of oocytes showing each defect. See 'Methods' for more details. 'N' and 'n' denote the number of oocytes and bivalents analysed, respectively.

The online version of this article includes the following source data and figure supplement(s) for figure 9:

**Figure supplement 1.** Strategy for the RNAi/rescue experiments using *bub-1* transgenes.

**Figure supplement 1—source data 1.** Gel scan images for data presented in *Figure 9—figure supplement 1*.

**Figure supplement 2.** Disruption of BUB-1- and CENP-C$^{HCP-4}$ STP motifs leads to severe meiotic defects.

CENP-C (203-IPTILE-208). This is relevant because it has been shown that PLK1 and B56 motifs tend to co-exist in close proximity (*Cordeiro et al., 2020*; *Singh et al., 2021*). This makes a putative cross-talk between PLK-1 and PP2A/B56 a worthy avenue to follow-up on our current findings.

Apparent lack of a CCAN network along with retention of crucial roles for PLK1 in species like *C. elegans* and *Drosophila melanogaster* suggests that the CENP-U pathway could have been replaced by other proteins. CENP-C is a good candidate, as the only CCAN component in these species. While we confirm that this is the case in *C. elegans*, it is interesting to note that putative STPs exist in sequences 176–178 and 266–268 in *D. melanogaster* CENP-C (UniProt #Q9VHP9). Interestingly, *D. melanogaster* Polo and CENP-C co-localise and this co-localisation increases by ectopic centromere generation through CENP-A (CID) over-expression (*Heun et al., 2006*).

Overall, our results advance our understanding on the roles played by PLK-1 during oocyte meiosis and provide a mechanistic understanding of PLK-1 targeting to meiotic chromosomes. The next step

will be to identify and characterise PLK-1 meiotic substrates to understand exactly how PLK-1 participates in each of its meiotic roles.

## Methods

### *C. elegans* strains and RNAi

Strains used in this study were maintained at 20° unless indicated otherwise. For a complete list of strains, please refer to *Supplementary file 1*.

For RNAi-mediated depletions, the targeting sequence for *bub-1* was 2353–2935 and for *hcp-4*, 967–2128, both from the first ATG codon. For double depletion, both sequences were cloned in the same vector. All sequences were inserted into L4440 using the NEBuilder HiFi DNA Assembly Master Mix (New England Biolabs) and transformed into DH5a bacteria. The purified plasmids were then transformed into HT115(DE3) bacteria (*Timmons et al., 2001*). RNAi clones were picked and grown overnight at 37°C in LB with 100 µg/ml ampicillin. Saturated cultures were diluted 1:100 and allowed to grow until reaching an OD600 of 0.8–1. Isopropyl-β-d-thiogalactopyranoside (IPTG) was added to a final concentration of 1 mM, and cultures were incubated for 1 hr at 37°C. Bacteria were then seeded onto NGM plates made with agarose and 1 mM IPTG and allowed to dry. L4 worms were then plated on RNAi plates and maintained at 20°C.

### CRISPR/Cas9

We used direct injection of in vitro-assembled Cas9-CRISPR RNA (crRNA) trans-activating crRNA (tracrRNA) ribonucleoprotein complexes (*Paix et al., 2017*; *Paix et al., 2015*).

For mutation of threonine 527 to alanine in *bub-1*, we used the following crRNA (+strand): CCCCGCACAAGGAGTTCATT and repair template (- strand): acttacTAATTTACTGAAAGTACT GCTGGTTGGAGCAACAAATACTGGAGCTTCCTGTTCGTGAGTGCTTTCCTCCTCTTTATTTC CGAAATATTCATCAATGTTGACtAAATGAACTCCTTGTGCGGGgGcactaGTGACGAAATTA CCACGAGACGGTTTGAAAAAGCCAAACTCGATTTCATCGTCATAAActaaaa
For mutation of threonine 163 to alanine in *hcp-4*, we used the following crRNA (- strand): TGAA ATATCAAGCGATCTCA and repair template (+ strand): taataaatctataatttcagAGTGGAAAAGCT GGATTAAGcTGcagtgCaCCCAAGAGCTCGAGTGATACGTCGATGAGGTCtTTGAGATCGCTTG ATATTTCACATGTCGTCAATACCGATC.

Each of the mixes was mixed with dpy-10 crRNA/repair template for screening (*Arribere et al., 2014*).

### Live imaging of oocytes

A detailed protocol for live imaging of *C. elegans* oocytes was used with minor modifications (*Laband et al., 2018*). Fertilized oocytes were dissected and mounted in 5 µl of L-15 blastomere culture medium (0.5 mg/mL inulin; 25 mM HEPES, pH 7.5 in 60% Leibowitz L-15 medium and 20% heat-inactivated FBS) on 24 × 40 mm #1.5 coverslips. In experiments where the PP1 analogue was used, 3IB-PP1 was diluted to the desired concentration in L-15 media from a 10 mM stock solution in ethanol. Control experiments were conducted with the equivalent dilution of ethanol. The incubation time while dissection took place was typically between 3 and 5 min, unless otherwise stated. Once dissection was performed and early oocytes identified using a stereomicroscope, a circle of Vaseline was laid around the sample, and a custom-made 24 × 40 mm plastic holder (with a centred window) was placed on top. The sample was imaged immediately using 488 nm and 561 nm laser lines. Live imaging was done using a CFI Plan Apochromat Lambda 60×/NA 1.4 oil objective mounted on a microscope (Nikon Eclipse Ti) equipped with a Prime 95B 22 mm camera (Photometrics), a spinning-disk head (CSU-X1; Yokogawa Electric Corporation). Acquisition parameters were controlled with NIS software (Nikon). For all live imaging experiments, partial projections are presented. All files were stored, classified, and managed using OMERO (*Allan et al., 2012*). Figures were prepared using OMERO.figure and assembled using Adobe Illustrator. Representative movies shown in the Supplementary material were assembled using Fiji/ImageJ (*Schindelin et al., 2012*) custom-made macros.

### Viability assay

Three different PP1 analogues were tested for their effect on embryo viability. Three worms were placed in NGM-coated wells of 24-well plates containing vehicle ('control') or the indicated concentration of

each analogue, in triplicate wells per condition. Once ≥200 embryos had been laid per well (with the exception of 50 µM analogue, where ≥100 embryos were analysed), adult worms were removed, and the exact number of embryos recorded. 72 hr later, viable worms were counted. In experiments where the PP1 analogue was used, 3IB-PP1 was diluted to the desired concentration in L-15 media from a 10 mM stock solution in ethanol. Control experiments were conducted with the equivalent dilution of ethanol. The incubation time while dissection took place was typically between 3 and 5 min, unless otherwise stated.

## Immunofluorescence

Worms were placed on 4 µl of M9 worm buffer in a poly-D-lysine (Sigma, P1024)-coated slide and a 24 × 24 cm coverslip was gently laid on top. Worms were dissected to release the embryos, and slides were placed on a metal block on dry ice for >10 min. The coverslip was then flicked off with a scalpel blade, and the samples were fixed in methanol at 20°C for 30 min. Secondary antibodies were donkey anti-sheep, goat anti-mouse, or goat anti-rabbit conjugated to Alexa Fluor 488, Alexa Fluor 594, and Alexa Fluor 647 (1:1000, Thermo Scientific). Donkey anti-mouse and donkey anti-rabbit conjugated secondary antibodies were obtained from Jackson ImmunoResearch. Embryos were mounted in ProLong Diamond antifade mountant (Thermo Scientific) with DAPI. Primary antibodies were α-PLK-1 (*Budirahardja and Gönczy, 2008*), α-HCP-4 (*Oegema et al., 2001*), and α-BUB-1, purified in house after immunisation of rabbits using the sequence in *Desai et al., 2003*.

## Sequence alignments

Sequences shown in *Figures 2C and 5B* were aligned with Clustal Omega (*Sievers et al., 2011*) and visualised with Jalview (*Waterhouse et al., 2009*).

## Protein purification

GST-BUB-1$^{190-628}$ and GST-HCP-4$^{1-214}$ proteins were expressed in *Escherichia coli* BL21 DE3 bacteria by diluting a saturated culture 1/100 in LB media supplemented with 35 µg/ml ampicillin and incubating at 37°C/200 rpm until $OD_{600}$ 0.6–0.8 was reached. IPTG was then added to a final concentration of 100 µM and cultures incubated at 20°C/200 rpm for 16–18 hr. The bacterial cultures were then centrifuged (20 min/6250 × $g$/4°C) and pellets resuspended in lysis buffer (50 mM Tris-HCl, pH 7.5; 150 mM NaCl; 0.5 mM TCEP, 1X Roche cOmplete protease inhibitors, EDTA free; 30–35 ml/l of culture). The cell suspension was then sonicated (2 min 40 s, 20 s on/40 s off) before centrifugation (45 min, 27,250 × $g$, 4°C) to remove insoluble material. GSH sepharose beads were washed with 10 column volumes (CV) of MilliQ water and equilibrated with 10 CV of binding buffer (50 mM Tris-HCl, pH 7.5, 150 mM NaCl, 0.5 mM TCEP) before filtered (0.22 µm PES filter) lysate was added and batch bound at 4°C for 1.5–2 hr. The beads were then collected in a column and washed with at least 10 CV of binding buffer before being transferred to a Falcon tube and incubated overnight with GST-tagged 3C protease. After cleavage, the beads were collected in a column and the flowthrough containing cleaved protein was concentrated in a Vivaspin centrifugal concentrator. The protein was then further purified by SEC in a Superdex 200 10/300 column (run in SEC buffer, see below) and concentrated before being flash frozen in liquid nitrogen and stored at –80°.

MBP-PLK1$^{PBD}$ (6xHIS-MBP tagged human PLK1$^{345-603}$, as per *Singh et al., 2021*) was expressed and purified in the same manner as the GST-tagged proteins with the following exceptions: filtered lysate was passed through a cobalt-NTA column and washed with at least 10 CV of binding buffer (see buffers below) before elution in 0.5 CV fractions. Fractions containing the protein were concentrated using Vivaspin centrifugal filters and further purified by SEC (Superdex 200 10/300) in SEC buffer. Lysis buffer (50 mM Tris-HCl, pH 7.5; 500 mM NaCl; 10 mM imidazole; 0.5 mM TCEP; 1X Roche cOmplete protease inhibitors, EDTA free; 50 ml/l of culture); binding buffer (50 mM Tris-HCl, pH 7.5; 500 mM NaCl; 10 mM imidazole; 0.5 mM TCEP); elution buffer (50 mM Tris-HCl, pH 7.5; 150 mM NaCl; 200 mM imidazole; 0.5 mM TCEP); and SEC buffer (50 mM Tris-HCl, pH 7.5, 150 mM NaCl, 0.5 mM TCEP).

## Kinase assays

Unless otherwise stated, kinase assays were conducted with 55 nM Cdk1:Cyclin B (Thermo Scientific) and/or 75 nM PLK-1, produced as described in *Tavernier et al., 2015*, at 30°C in kinase buffer: 50 mM Tris-HCl, pH 7.5; 1 mM ATP; 10 mM $MgCl_2$; 0.5 mM TCEP; 0.1 mM EDTA. For the assays in *Figure 2E*

*and F* and *Figure 2—figure supplement 1*, 0.4 µg/µl (8.2 µM) of BUB-1$^{190-628}$ substrate was used and incubated under the above conditions for up to 1 hr. The assays in *Figure 5* were conducted in the above conditions for up to 1 hr with a substrate concentration of 0.2 µg/µl (8.2 µM). The assay in *Figure 2D* was conducted with 165 nM Cdk1:Cyclin B and 170 nM PLK-1 and incubated for 30 min at 37°C with the following buffer: 40 mM Tris-HCl, pH 7.5; 100 µM ATP, 10 mM MgCl$_2$. BUB-1$^{190-628}$ concentration was 0.2 µg/µl (4.1 µM).

In all assays, aliquots of protein at the indicated time points were immediately added to an equal volume of 2X LDS buffer (Thermo) and incubated at 70° for 15 min. Assays were assessed by SDS-PAGE combined with ProQ diamond phosphoprotein and Coomassie staining.

## In vitro binding assays

BUB-1$^{190-628}$ or CENP-C$^{HCP-4(1-214)}$ recombinant proteins were incubated for 50–60 min at 30°C in the presence or absence of ~100 nM Cdk1:Cyclin B in kinase buffer (see 'Kinase assay' section of 'Methods'). The respective proteins were then incubated for 50 min on ice with MBP-PLK1$^{PBD}$ at a concentration of 20 µM in SEC buffer (50 mM Tris-HCl, pH 7.5; 150 mM NaCl; 0.5 mM TCEP). Assays were then centrifuged (13.3 k rpm, 10 min, 4°C) before being loaded onto a Superdex 200 10/300 SEC column. In all assays, 0.2 ml fractions were collected and selected fractions were assessed by SDS-PAGE with Coomassie staining. More specifically, for the binding assays with BUB-1$^{190-628}$ in *Figure 3A and C*, 200 µl of binding assay was loaded into a 0.5 ml loop and loaded onto the column with 2 ml of SEC buffer. For the CENP-C$^{HCP-4(1-214)}$ assays in *Figure 6A and C* 100 µl of assay was loaded onto a 100 µl loop and loaded onto the column with 0.4 ml of SEC buffer.

## Size-exclusion chromatography multi-angle light scattering (SEC-MALS)

An Agilent 1260 Infinity II HPLC system was used to inject 50 µl of protein (ranging from 0.5 to 2.5 mg/ml) onto a Superdex 200 10/300 gel filtration column (Cytiva) coupled to a Wyatt miniDawn multi-angle light scattering system and Wyatt Optilab differential refractive index (dRI) detector. A dn/dc value of 0.185 was used, UV extinction coefficients were calculated from RI and molecular weights were calculated using Zimm plot extrapolation. Astra v8.1 was used for data analysis. The following buffer was used in all experiments: 50 mM Tris-HCl, pH 7.5, 150 mM NaCl, 0.5 mM TCEP.

## Western blot

In vitro kinase assays were conducted as described above and separated by SDS-PAGE on 4–12% Bis-Tris gels (Thermo) with ~20 ng of protein loaded per well for the α-BUB-1 or α-HCP-4 blots and either 80 ng (BUB-1$^{190-628}$) or 75 ng (CENP-C$^{HCP-4(1-214)}$) loaded per lane for the phosphospecific antibodies. Transfer was conducted for 90 min at 90 V in transfer buffer (25 mM Tris, 200 mM glycine, 20% methanol) before membrane was washed in PBS-T (PBS with 0.1% Tween-20) and blocked with Intercept PBS blocking buffer (LI-COR). Membranes were washed in PBS-T before incubation with primary antibodies overnight at 4°C. For the peptide competition blots, peptide or phosphopeptides were added to a final concentration of 1 µM alongside the primary antibodies. The next day, membranes were washed in PBS-T before incubation with LI-COR secondary antibodies (1:25,000) in 5% milk for 1 hr. Membranes were then washed once more in PBS-T before scanning in a LI-COR Odyssey CLx.

Worm extract western blots (*Figure 9—figure supplement 1*) worms were grown and transferred to RNAi plates as described above. 50 worms per lane were picked and washed with M9 + 0.1% triton (x2) and M9 (x1). LDS buffer was then added to a final concentration of 1×, and samples were heated (70°C, 5 min), sonicated (Diagenode cell disruptor, high power, 10 cycles, 30 s ON/30 s OFF), and heated again (70°C, 10 min) before western blots were conducted following the protocol described above.

## Fluorescence polarisation

Fluorescence polarisation assays were conducted using FITC-labelled peptides (peptide sequences indicated in figures). For all assays, a 1:2 dilution series of MBP-PLK1$^{PBD}$ was conducted in FP buffer (50 mM Tris-HCl, pH 7.5, 150 mM NaCl, 0.5 mM TCEP) with a constant concentration of 100 nM FITC-labelled peptide. Assays were left for 20–60 min before being centrifuged and 10 µl of each concentration was loaded onto a black 384 well plate (Greiner) in triplicate. Plates were then centrifuged (2k rpm, 2 min) before being analysed in a PheraStar FS (BMG Labtech) under the following conditions:

excitation: 485 nm; emission: 520 nm, 25 °C; settling time 0.2 s, 50 flashes per well, gain and focal height automatically adjusted prior to measurement. Mean mP for each triplicate was then used as the value for each independent experiment. Data represent the average of three independent experiments with $K_D$ values being estimated using GraphPad Prism 9 'Nonlinear regression, One Site – Total' equation. The uncertainty indicated represents the standard deviation of the calculated $K_D$ from each independent replicate.

## Phenotype analysis

We defined misalignment as metaphase defects. Alignment was counted within the five frames (1 min) before anaphase onset, which was the frame prior to the detection of two separating chromosome masses. When one bivalent was misaligned (either in angle or distance to metaphase plate), it was recorded as mild metaphase defect. When two or more bivalents were misaligned, this was considered a severe metaphase defect. The anaphase phenotype was assess during segregation. If lagging chromosome were detected during chromosome segregation but this was resolved before polar body extrusion, it was scored as mild anaphase defect. If we could not detect two segregating masses of chromosomes or if these masses differed in size/intensity, it was quantified as a severe anaphase defect. Polar body defect was recorded when no polar body was extruded or if all of the maternal DNA content was extruded as a polar body, with no maternal DNA remaining in the cytoplasm. Graphs were prepared using GraphPad Prism 9.0.

## Acknowledgements

We thank Arshad Desai for sharing strains and antibodies and for helpful discussions and sharing results prior to publication; Pierre Gönczy for the PLK-1 antibody; Lionel Pintard for providing PLK-1 for the initial experiments and the plasmid for expression and purification; Bruce Bowerman for the GFP::ASPM-1 strain; Satpal Virdee for advice on fluorescence polarisation experiments, and Vincent Postis for guidance with the SEC-MALS experiments. We also thank Ron Hay for comments on the manuscript. This work was supported by a Career Development Award from the Medical Research Council (grant MR/R008574/1) and an ISSF grant funded by the Wellcome Trust (105606/Z/14/Z). ST is funded by a Medical Research Council Doctoral Training Programme. DKC is supported by a Sir Henry Dale Fellowship from the Wellcome Trust (208833). JH received support from NSF Graduate Research Fellowship (grant #1650112) and from a grant from the NIH to A Desai (GM074215). We acknowledge the FingerPrints Proteomics Facility and the Dundee Imaging Facility, which are supported by a 'Wellcome Trust Technology Platform' award (097945/B/11/Z) and the Tissue Imaging Facility, funded by a Wellcome Trust award (101468/Z/13/Z). Some nematode strains were provided by the CGC, which is funded by the NIH Office of Research Infrastructure Programs (P40 OD010440).

## Additional information

### Competing interests

Federico Pelisch: Reviewing editor, *eLife*. The other authors declare that no competing interests exist.

### Funding

| Funder | Grant reference number | Author |
| --- | --- | --- |
| Medical Research Council | MR/R008574/1 | Laura Bel Borja Flavie Soubigou Federico Pelisch |
| Wellcome Trust | 208833 | Dhanya K Cheerambathur |
| National Science Foundation | 1650112 | Jack Houston |
| National Institutes of Health | GM074215 | Jack Houston |
| Wellcome Trust | 105606/Z/14/Z | Federico Pelisch |

| Funder | Grant reference number | Author |
|---|---|---|
| Medical Research Council | Doctoral Training Programme | Samuel JP Taylor |

The funders had no role in study design, data collection and interpretation, or the decision to submit the work for publication. For the purpose of Open Access, the authors have applied a CC BY public copyright license to any Author Accepted Manuscript version arising from this submission.

## Author contributions
Samuel JP Taylor, Investigation, Visualization, Methodology, Writing - original draft, Writing - review and editing; Laura Bel Borja, Flavie Soubigou, Formal analysis, Investigation, Visualization; Jack Houston, Resources; Dhanya K Cheerambathur, Resources, Writing - review and editing; Federico Pelisch, Conceptualization, Formal analysis, Supervision, Funding acquisition, Investigation, Methodology, Writing - original draft, Project administration, Writing - review and editing

## Author ORCIDs
Samuel JP Taylor ⓘ http://orcid.org/0000-0002-0654-619X
Laura Bel Borja ⓘ http://orcid.org/0000-0002-8381-934X
Federico Pelisch ⓘ http://orcid.org/0000-0003-4575-1492

## Decision letter and Author response
Decision letter https://doi.org/10.7554/eLife.84057.sa1
Author response https://doi.org/10.7554/eLife.84057.sa2

## Additional files

### Supplementary files
• Supplementary file 1. List of *C. elegans* strains used in this study. The table details the fluorescent markers and/or mutations, strain name, genotype, and source.

• MDAR checklist

### Data availability
A supporting file containing all the information for the graphs presented in the manuscript has been added.

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
