## [Editor Report]

This study represents an important advance in our understanding of how female meiosis is regulated. By combining biochemistry with some beautiful cell biology, the authors identify and increase our understanding of how chromosome behaviour is controlled in mitosis. The data are convincing and the paper will be of interest to researchers in the mitosis and meiosis fields.

---

## [Decision Letter]

**Decision letter after peer review:**

Thank you for submitting your article "BUB-1 and CENP-C Recruit PLK-1 to Control Chromosome Alignment and Segregation During Meiosis I in *C. elegans* Oocytes" for consideration by *eLife*. Your article has been reviewed by 3 peer reviewers, and the evaluation has been overseen by a Reviewing Editor and Anna Akhmanova as the Senior Editor. The reviewers have opted to remain anonymous.

Essential revisions:

1) Please generate and characterise a double hcp-4T163 bub-1527A mutant, as suggested by reviewers 2 and 3. The localization of a few RC/chromosomal proteins should be analysed in this mutant, to determine if they are delocalized (this analysis can be limited to a representative subset of proteins, for example KLP-19 and MCAK). If you find that other proteins are delocalized in this mutant (in addition to PLK-1), this should be discussed in the interpretation of the phenotypes.

2) Please report the number of samples and number of experiments for each figure.

3) Please perform the control experiment of treating wild type oocytes with 10uM 3-IB-PP1.

4) All the other reviewers' comments can be addressed by revising the text.

*Reviewer #1 (Recommendations for the authors):*

– For some of the figures, the images are very small – it would be helpful for the reader if the panels were resized/reconfigured to maximize the images sizes. For example, in Figure 1, the diagram of the bivalent (Figure 1A) is quite large, but the images in panel 1G are so small that they are hard to see, even when I increase the size of the figure on my screen. (Similar issues are in some of the other figures, e.g. Figure 3D, 6B, 6E). I recommend revisiting some of the figures with this comment in mind, minimizing white space in the figure and re-organizing the panels to maximize image size.

– Figure 1B: it is hard to see the midbivalent localization in the live images (the regions magenta arrows are pointing at) – showing zooms of this region would make this clearer (similar to what is shown for the fixed images in Figure 1D).

– Figure 1F: if these are stills from live imaging, timestamps should be added.

– Figure 1H: it would be helpful to label on this panel somewhere what concentration of inhibitor was used.

– Figure 1I: It would be helpful to add a label to the figure denoting how long the oocytes were treated with 3IB-PP1 (I couldn't find this information in the figure legend either). Also, the legend states that early metaphase spindles were chosen for each condition, but since the spindle is disrupted at the 10uM concentration, it is not clear how it would be possible to stage this spindle (so it should not be called early metaphase).

– Figure 2A/2B: the midbivalent localization is not clear – PLK-1 looks diffuse all over the chromosome, not specifically localized to the midbivalent. The imaging in Figure 3A (for a different experiment) is clearer – could the images in Figure 2A/2B be similarly improved to make the data more convincing?

– Figure 2C: the domain diagram is very helpful. It might be nice to also mark on this diagram the region expressed in the in vitro analysis (190-628) so the reader can easily see which domains this represents.

– In describing the analysis shown in Figure 2 —figure supplement 1, it would be helpful to explain to the reader that CDK-1 and PLK-1 can both phosphorylate BUB-1 even when T527 is mutated to alanine. This is in the figure, but never mentioned – I think it might help the reader understand the data better if you make it clear that these kinases can phosphorylate BUB-1 on other sites (not just T527), so they are not confused by the ProQ band in these lanes (and in Figure 2E).

– Figure 2F, 2G: what is the dashed line on the gels? Does it represent a specific fraction or elution volume, or is it just there as a comparison between gels? Explain this in figure legend for clarity.

– Figure 3C: the top row is labeled PLK-1 but I think it is supposed to be BUB-1.

– Throughout the manuscript, sometimes PLK-1 is denoted with a dash (PLK-1) and sometimes without (PLK1). This was particularly prominent in the sections describing the in vitro data (in the results and the figures; for example compare the labeling in Figure 5B and 5C to Figure 5 —figure supplement 1). This notation should be standardized throughout the paper and figures (a dash should always be used when describing *C. elegans* PLK-1).

– Figure 5G: the top row is labeled PLK-1 but I think it is supposed to be BUB-1.

*Reviewer #2 (Recommendations for the authors):*

1) Phenotypic analysis of the double hcp-4T163 bub-1527A mutant oocytes would be interesting to look at, as this genotype should "surgically" prevent PLK-1 recruitment without the need to entirely deplete BUB-1 and HCP-4 functions which might well have other roles beyond promoting PLK-1 recruitment, particularly BUB-1.

2) In the introduction page 3 line 51, please mention that PLK1 also localizes to the nuclear envelope in prophase (Linder et al. Dev Cell 2017).

3) Page 8 lane 138, I do not see any evidence indicating that 3IB-PP1 is more specific. While Figure 1-Figure S2 shows it is more potent, the evidence in the figure does not really address its specificity. Furthermore, for this figure the authors should indicate the number of embryos analyzed. It is not clear based on the figure and the figure legends.

4) The authors should explain how the graphs presented in Figure 3E, 6E have been generated. How many oocytes were analyzed? How many different experiments?

5) Several figures are lacking scale bars. (Figure 1D, 1G, 3A, 3C).

6) Molecular weight markers (MW) are missing on all the figures showing SDS-PAGE.

*Reviewer #3 (Recommendations for the authors):*

My comment on the manuscript is that I was not able to find any number (how many samples were analyzed each experiment, how many times the experiments were performed).

---

## [Author Response]

Essential revisions:1) Please generate and characterise a double hcp-4T163 bub-1527A mutant, as suggested by reviewers 2 and 3. The localization of a few RC/chromosomal proteins should be analysed in this mutant, to determine if they are delocalized (this analysis can be limited to a representative subset of proteins, for example KLP-19 and MCAK). If you find that other proteins are delocalized in this mutant (in addition to PLK-1), this should be discussed in the interpretation of the phenotypes.

We acknowledge the key point of generating the double point mutant. To achieve this, we used an RNAi/rescue approach, whereby we depleted endogenous BUB-1 with RNAi and expressed RNAi-resistant wild type or T527A BUB-1 transgenes. The new Figure 9 now shows that:

1) wild type but not T527A BUB-1 rescues PLK-1 localisation. In the wild type *hcp-4* background this is observed in the midbivalent PLK-1. In the *hcp-4^T163A^* background, BUB-1 is sufficient to drive kinetochore (see below for more discussion on kinetochore localisation) and midbivalent PLK-1 localisation. Overall, PLK-1 localisation to meiotic chromosomes relies almost entirely on the polo-docking ‘STP’ motifs in BUB-1 and HCP-4.

2) *hcp-4^T163A^/bub-1^T527A^* oocytes display meiotic defects similar to *hcp-4^T163A^/bub-1RNAi* and *hcp-4(RNAi)/bub-1^T527A^*, confirming that BUB-1 and HCP-4 target PLK-1 through their STP motifs to control meiotic chromosome segregation.

This paper focuses on identifying the specific meiotic events PLK-1 plays a role in and characterising its targeting mechanism. We are following on this work to understand what proteins are regulated by PLK-1 in different chromosome domains and how this relates to the observed phenotypes. For this paper, a single Thr residue within an STP motif in a largely disordered region is far more specific than depleting HCP-4 or BUB-1, making it likely that these effects are mediated through PLK-1 targeting.

2) Please report the number of samples and number of experiments for each figure.

We have now included all the relevant information about number of independent experiments and sample size.

3) Please perform the control experiment of treating wild type oocytes with 10uM 3-IB-PP1.

This control has now been included in Figure 1—figure supplement 3.

4) All the other reviewers' comments can be addressed by revising the text.

All comments have been addressed.

Reviewer #1 (Recommendations for the authors):– For some of the figures, the images are very small – it would be helpful for the reader if the panels were resized/reconfigured to maximize the images sizes. For example, in Figure 1, the diagram of the bivalent (Figure 1A) is quite large, but the images in panel 1G are so small that they are hard to see, even when I increase the size of the figure on my screen. (Similar issues are in some of the other figures, e.g. Figure 3D, 6B, 6E). I recommend revisiting some of the figures with this comment in mind, minimizing white space in the figure and re-organizing the panels to maximize image size.

We have re-organised the panels in Figures 1, 3, and 6 to increase the size of the relevant data. Additionally, we made sure these are high resolution to allow for proper zooming into each panel.

– Figure 1B: it is hard to see the midbivalent localization in the live images (the regions magenta arrows are pointing at) – showing zooms of this region would make this clearer (similar to what is shown for the fixed images in Figure 1D).

We have re-organised Figure 1 and the new panel C which shows PLK-1 localisation in live oocytes now also displays zooms of individual bivalent.

– Figure 1F: if these are stills from live imaging, timestamps should be added.

Thanks for pointing this out. Timestamps were left out by mistake and have now been added.

– Figure 1H: it would be helpful to label on this panel somewhere what concentration of inhibitor was used.

We have added the PP1 analogue concentration, and the graph is now Figure 1G.

– Figure 1I: It would be helpful to add a label to the figure denoting how long the oocytes were treated with 3IB-PP1 (I couldn't find this information in the figure legend either). Also, the legend states that early metaphase spindles were chosen for each condition, but since the spindle is disrupted at the 10uM concentration, it is not clear how it would be possible to stage this spindle (so it should not be called early metaphase).

We and others have observed that in situations that disrupt spindle morphology, anaphase onset can be noticed by chromosome coming together while the tubulin signal area is reduced, and its signal becomes more intense. This is the case after PP2A depletion (Bel Borja et al. 2022 – PMID: 33355089), KLP-18 depletion (Muscat et al. 2015 – PMID: 26026148), KLP-7 depletion (Gigant et al. 2017 – PMID: 28289130; Connolly et al. 2015 – PMID: 26370499). While this is not of course an accurate way to describe metaphase, it is as close as we can get. We have now made sure we mention this in the legend.

– Figure 2A/2B: the midbivalent localization is not clear – PLK-1 looks diffuse all over the chromosome, not specifically localized to the midbivalent. The imaging in Figure 3A (for a different experiment) is clearer – could the images in Figure 2A/2B be similarly improved to make the data more convincing?

The new Figure 2A makes the midbivalent PLK-1 localisation clearer than before.

– Figure 2C: the domain diagram is very helpful. It might be nice to also mark on this diagram the region expressed in the in vitro analysis (190-628) so the reader can easily see which domains this represents.

Thanks for this suggestion. A schematic representation of the BUB-1^190-628^ and CENP-C^HCP-4(1-214)^ proteins is now included in figures 2D,F and 5E.

– In describing the analysis shown in Figure 2 —figure supplement 1, it would be helpful to explain to the reader that CDK-1 and PLK-1 can both phosphorylate BUB-1 even when T527 is mutated to alanine. This is in the figure, but never mentioned – I think it might help the reader understand the data better if you make it clear that these kinases can phosphorylate BUB-1 on other sites (not just T527), so they are not confused by the ProQ band in these lanes (and in Figure 2E).

We have now adapted the wording and added western blots with phospho-specific antibodies for the two STP motifs (phT527 BUB-1 and phT163 HCP-4) which makes this distinction clearer.

– Figure 2F, 2G: what is the dashed line on the gels? Does it represent a specific fraction or elution volume, or is it just there as a comparison between gels? Explain this in figure legend for clarity.

Thanks for this comment – the dashed lines were previously included to aid comparison of the different gels within each experiment. On reflection, this does not make analysis of the result any clearer, so we have removed them from all gels.

– Figure 3C: the top row is labeled PLK-1 but I think it is supposed to be BUB-1.

This has now been corrected. Thanks for pointing this out.

– Throughout the manuscript, sometimes PLK-1 is denoted with a dash (PLK-1) and sometimes without (PLK1). This was particularly prominent in the sections describing the in vitro data (in the results and the figures; for example compare the labeling in Figure 5B and 5C to Figure 5 —figure supplement 1). This notation should be standardized throughout the paper and figures (a dash should always be used when describing *C. elegans* PLK-1).

Thanks for the comment. We have adapted the text so PLK1 is used for the information that specifically comes from mammalian systems while PLK-1 is used when discussing the protein in general and when specifically referring to the *C. elegans* protein.

– Figure 5G: the top row is labeled PLK-1 but I think it is supposed to be BUB-1.

This has now been corrected. Thanks for pointing this out.

Reviewer #2 (Recommendations for the authors):1) Phenotypic analysis of the double hcp-4T163 bub-1527A mutant oocytes would be interesting to look at, as this genotype should "surgically" prevent PLK-1 recruitment without the need to entirely deplete BUB-1 and HCP-4 functions which might well have other roles beyond promoting PLK-1 recruitment, particularly BUB-1.

We have performed these experiments, which are now presented in Figure 9.

2) In the introduction page 3 line 51, please mention that PLK1 also localizes to the nuclear envelope in prophase (Linder et al. Dev Cell 2017).

We added the reference.

3) Page 8 lane 138, I do not see any evidence indicating that 3IB-PP1 is more specific. While Figure 1-Figure S2 shows it is more potent, the evidence in the figure does not really address its specificity. Furthermore, for this figure the authors should indicate the number of embryos analysed. It is not clear based on the figure and the figure legends.

We agree with this comment, the text was not phrased well and has been adapted accordingly. The number of embryos analysed for viability assays is now stated in the ‘Methods’ section.

4) The authors should explain how the graphs presented in Figure 3E, 6E have been generated. How many oocytes were analyzed? How many different experiments?

We have now included in the Results section a more thorough explanation of how we generated these graphs (which was only present in the Methods in the previous version). Additionally, we have included the number of oocytes analysed for each condition, with all experiments having been repeated ³3 times.

5) Several figures are lacking scale bars. (Figure 1D, 1G, 3A, 3C).

Scale bars have been added.

6) Molecular weight markers (MW) are missing on all the figures showing SDS-PAGE.

MW markers have been added.

Reviewer #3 (Recommendations for the authors):My comment on the manuscript is that I was not able to find any number (how many samples were analyzed each experiment, how many times the experiments were performed).

We now report the number of oocytes (N) and bivalents (n) analysed for localisation analysis. For phenotype analysis, we have included the number of oocytes analysed in each condition, with all experiments having been repeated ³3 times.